# UNIFYING DISENTANGLED REPRESENTATION LEARNING WITH COMPOSITIONAL BIAS

## ABSTRACT

Existing disentangled representation learning methods rely on inductive biases tailored for specific factors of variation (e.g., attributes or objects). However, these biases are incompatible with other classes of factors, limiting their applicability for disentangling general factors of variation. In this paper, we propose a unified framework for disentangled representation learning, accommodating both attribute and object disentanglement. To this end, we reformulate disentangled representation learning as maximizing the compositionality of the latents. Specifically, we randomly *mix* two latent representations from distinct images and maximize the likelihood of the resulting composite image. Under this general framework, we demonstrate that adjusting the strategy for mixing between two latent representations allows us to capture either attributes or objects within a single framework. To derive appropriate mixing strategies, we analyze the compositional structures of both attributes and objects, then incorporate these structures into their respective mixing strategies. Our evaluations show that our method surpasses or is comparable to state-of-the-art baselines such as DisDiff (Yang et al., 2023) in attribute disentanglement (DCI, FactorVAE scores), and LSD (Jiang et al., 2023) and L2C (Jung et al., 2024) in object property prediction tasks for object disentanglement.

## 1  INTRODUCTION

Understanding the underlying structure of data is crucial for building robust and interpretable machine learning models. In particular, by perceiving the world through compositional concepts, unseen data can be decomposed into simpler, more interpretable components. This approach dramatically improves data efficiency of learning, as unseen data can be explained as combinations of already learned concepts (Lake et al., 2017; Kuo et al., 2021). In this context, disentangled representation learning (Higgins et al., 2018; Bengio & LeCun, 2007) aims to decompose the data into its underlying factors of variation. As Locatello et al. (2019) theoretically prove that disentangled representation learning cannot be achieved without inductive biases or direct supervision, the field has focused on designing appropriate inductive biases to disentangle desirable factors in an unsupervised manner.

Attribute and object disentanglement are two of the most common tasks in disentangled representation learning. We commonly refer to attributes as properties shared globally across the entire scene (e.g., color, lighting, or style), while objects are distinct spatial components within a scene (e.g., individual entities). Attribute disentanglement (Burgess et al., 2017; Chen et al., 2016; Kim & Mnih, 2018; Chen et al., 2018; Ren et al., 2022) aims to isolate various features or properties of the data. Disentanglement between latent variables is often encouraged by additional regularization terms, such as minimizing Total Correlation in VAEs (Burgess et al., 2017; Kim & Mnih, 2018; Chen et al., 2018), maximizing mutual information between latents and images Chen et al. (2016); Lin et al. (2020b), or minimizing mutual information between vector-wise latents (Yang et al., 2023). On the other hand, object-centric learning focuses on decomposing scenes into individual objects (Burgess et al., 2019; Greff et al., 2019; Engelcke et al., 2020; Locatello et al., 2020; Jiang et al., 2023). These methods rely on a spatial exclusiveness bias, where each pixel in an image must correspond to a unique object, implemented within model architectures such as spatial-attention masks (Burgess et al., 2019; Engelcke et al., 2020), pixel-mixture decoders (Greff et al., 2019), or slot attention (Locatello et al., 2020).

While both attribute and object disentanglement share the common goal of identifying underlying factors of variation, the aforementioned inductive biases are crafted specific to their respective target

factors and are incompatible with each other. Moreover, relying on these inductive biases may limit their extension to disentangling general factors of variation or scenarios that involve both attributes and objects in a scene. This challenge motivates us to develop a unified inductive bias capable of capturing a broader range of factors of variation. In this paper, we present a unified framework for disentangled representation learning that supports both attribute and object disentanglement. Inspired by the fact that the goal of disentangled representation learning is to achieve combinatorial generalization, we formulate disentangled representation learning as the process of maximizing compositionality and carefully design the composition rule that ensures valid combinations of latents. Specifically, given two sets of latent representations from different images, we construct a composite representation by exchanging random subsets of latents and maximize the likelihood of the resulting composite image. By analyzing the compositional structures of attributes and objects, we derive specific mixing strategies that enable valid combinations for effective attribute and object disentanglement. Unlike previous methods, which introduce inductive biases specifically tailored to either attribute or object disentanglement and are not compatible with both, our framework can handle both types of factors. Our experiments demonstrate that our framework effectively disentangles both attributes and objects by simply adjusting the mixing strategy, without altering model architectures or objective functions.

Our contributions are as follows: **(1)** We present a unified framework for disentangled representation learning that effectively addresses both attribute and object disentanglement. **(2)** We derive simple yet effective mixing strategies for disentangling attributes or objects, drawing from their underlying compositional structures. **(3)** We compare our methods with strong baselines specifically designed for disentangled representation learning and object-centric learning and verify that our method can achieve comparable or even superior performance to the baselines.

## 2 BACKGROUND : DISENTANGLED REPRESENTATION LEARNING

In this section, we briefly review the two main streams of disentangled representation learning: attribute and object disentanglement. We discuss how previous methods have achieved disentanglement and why they are incompatible with each other. More in-depth discussions on related works are presented in Appendix A.1.

**Attribute disentanglement**    In attribute disentanglement, scenes are assumed to consist of a fixed number of random variables (Kim & Mnih, 2018). Typical approaches aim to discover independent latent variables by designing objective functions that promote their statistical independence. For instance, (Burgess et al., 2017; Kim & Mnih, 2018; Chen et al., 2018) use Total Correlation (Watanabe, 1960) within the VAE framework to assess independence between latent dimensions. Alternatively, (Lin et al., 2020b; Ren et al., 2022) introduce contrastive regularization, encouraging variations in each latent variable to produce distinct changes in the output space of GANs. Recently, Yang et al. (2023) proposed minimizing the upper bound of mutual information between latent variables. These *information-theoretic objectives* are suited for scenarios where each data is composed of a fixed set of factors, with each latent variable corresponding to a specific factor. However, when this assumption is violated, defining and directly measuring independence between latent variables becomes non-trivial. For example, in object-centric scenes, the same objects can appear in different spatial locations, complicating the definition of independence metrics for object representations.

**Object disentanglement**    In object-centric learning, random variables are assumed to be independent but share a generative mechanism, such that different orderings of the latents still produce identical images (Greff et al., 2019). Since measuring independence between object representations is challenging, object-centric approaches use architectural biases to promote independence indirectly. Early methods implemented spatial exclusiveness through decoders that render each latent into pairs of a RGB image and a mask, blending them to form the final output (Burgess et al., 2019; Greff et al., 2019; Engelcke et al., 2020; Lin et al., 2020a). Each mask corresponds to a distinct region, inducing spatial exclusivity among the latents. Slot attention (Locatello et al., 2020) adopts a spatially exclusive mechanism within the encoder, where each latent (slot) exclusively binds to spatial locations in the input images. These *spatial exclusive biases* constrain each latent to bind to non-overlapping spatial regions, and the auto-encoding objective encourages the encoder to cluster spatially correlated pixels. While these biases facilitate the disentangling of spatial factors, they restrict the ability to disentangle non-spatial factors like attributes.

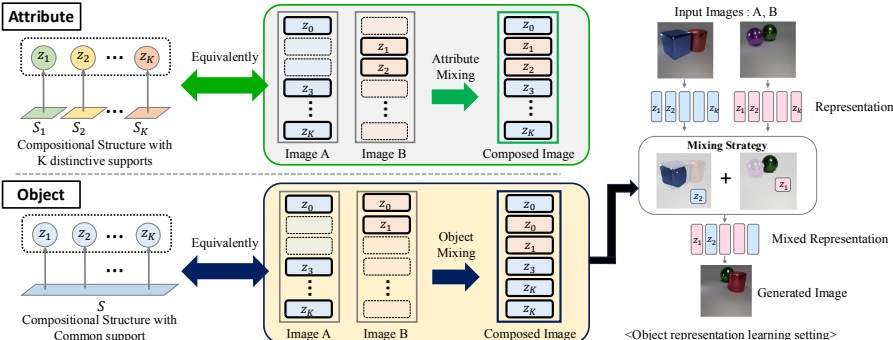

Figure 1: Overview of our method. We introduce a unified framework for disentangled representation learning, which is compatible with both attribute and object disentanglement. We formulate our framework as randomly composing latents from two distinct images, and maximizing the likelihood of resulting composite images (Section 3.1). To disentangle attributes and objects within this framework, we devise two specific mixing strategies to properly reflect their compositional structures (Section 3.2). Finally, we maximize the likelihood of composite images and ensure compositional consistency (Section 3.3). Note that the figure illustrates a specific example for object mixing strategy.

In summary, we identify two distinct inductive biases that promote independence between latent variables, either directly or indirectly. As these biases are tailored specifically to each class of factors of variation (*i.e.*, attributes and objects), they are not only incompatible with each other but also challenging to extend to disentangling general factors of variation. This challenge motivates us to seek a unified approach that can accommodate both attribute and object disentanglement.

# 3 UNIFYING DISENTANGLED REPRESENTATION LEARNING WITH COMPOSITIONAL BIAS

Our goal is to develop a unified framework for disentangled representation learning, which is compatible with both attribute and object disentanglement. In the following sections, we illustrate the overall framework to handle both attribute and object disentanglement (Section 3.1) and how we derive our new inductive bias from the different compositional structures of each factor of variation (Section 3.2). Finally, we demonstrate how we design the learning objectives to instantiate this general framework (Section 3.3). Figure 1 summarizes the overall framework of our method.

## 3.1 UNIFIED FRAMEWORK FOR LEARNING DISENTANGLED REPRESENTATION

Disentangled representation learning aims to represent an image $\mathbf{x} \in \mathbb{R}^{H \times W \times C}$ into a set of $K$ latent representations $\mathbf{z} = \{\mathbf{z}_i\}_{i=1}^{K}$, where each latent $\mathbf{z}_i \in \mathbb{R}^d$ is expected to capture independent factors of variation. Previous approaches achieve this goal by utilizing specific assumptions about the latent representations, such as statistical independence (Kim & Mnih, 2018; Chen et al., 2018) or spatial exclusiveness (Greff et al., 2019; Locatello et al., 2020) between the latent variables. Such assumptions are specific to the type of factors of variation (*e.g.*, attributes or objects) and imposed by tailored architecture or regularization, making them incompatible with different types of disentanglement.

Instead, we propose to employ the maximization of *compositionality* in the representation as a general objective for disentanglement learning while instantiating various disentanglement structures by controlling *only* the composition operator. To this end, we follow (Jung et al., 2024) by randomly composing latent representations from two images and maximizing the likelihood of the resulting composite image. Specifically, given two images $\mathbf{x}^1, \mathbf{x}^2 \sim p(\mathbf{x})$ and their representations $\mathbf{z}^1, \mathbf{z}^2 \in \mathbb{R}^{K \times d}$, respectively, we produce their composite representation $\mathbf{z}^c$ by some composition operator. Then, we decode $\mathbf{z}^c$ into a composite image $\mathbf{x}^c$ and maximize its likelihood $p(\mathbf{x}^c)$ to ensure the production of realistic images by:

$$\theta^* = \arg\max_{\theta} p(\mathbf{x}^c) = \arg\max_{\theta} p(D_\phi(\pi(\mathbf{z}^1, \mathbf{z}^2))) = \arg\max_{\theta} p(D_\phi(\pi(E_\theta(\mathbf{x}^1), E_\theta(\mathbf{x}^2)))), \quad (1)$$

where $E_\theta, D_\phi$ denote an encoder and a decoder, respectively. $\pi(\cdot, \cdot)$ represents a mixing operation between two sequences of representations such that $\pi(\mathbf{z}^1, \mathbf{z}^2) = \{\mathbf{z}_i^c \mid \mathbf{z}_i^c = \mathbf{z}_{\sigma_i}^{r_i}, \ i \in \{1, \ldots, K\}\}$, where $r_i \in \{1, 2\}$ indicates whether the $i$-th element is selected from $\mathbf{z}^1$ or $\mathbf{z}^2$, and $\sigma_i \in \{1, \ldots, K\}$ is an index that determines the order. Note that this formulation does not impose any assumptions specific to the factors of variation on the latent space. While (Jung et al., 2024) rely on architectural biases (e.g., slot attention) to improve object representations by maximizing compositionality, our work primarily explores how different types of factors can be disentangled by carefully designing the mixing operator $\pi(\cdot, \cdot)$. In the following section, we will demonstrate how we derive a specific mixing operator $\pi(\cdot, \cdot)$—referred to as the *mixing strategy*—for two representative examples of factors of variation: attributes and objects.

## 3.2 MIXING STRATEGY FOR REFLECTING THE COMPOSITIONAL STRUCTURE

The mixing strategy is defined to produce a random composition between two sequences of latent representations: $\mathbf{z}^1, \mathbf{z}^2$. It is important to note that not all random compositions result in valid outcomes. For instance, when we mix the ground-truth factors of face attributes, the composition having two noses becomes an invalid composition. This is because ground-truth factors follow a certain structure to be composed into a complete image, which we refer to as the *compositional structure* of factors of variation. Therefore, we characterize the compositional structure of each factor of variation, and derive a corresponding mixing strategy. We start by defining disentangled representation, following (Roth et al., 2023).

**Definition 1 (Disentanglement with a factorized support)** *Let us denote the support of $p(\mathbf{x})$ as $\mathcal{S}(p(\mathbf{x})) = \{\mathbf{x} | p(\mathbf{x}) > 0\}$. Given a sequence of random vectors $\mathbf{z} = \{\mathbf{z}_i\}_{i=1}^K$, $\mathbf{z}$ is disentangled with a factorized support if $\mathcal{S}(p(\mathbf{z})) = \mathcal{S}(p(\mathbf{z}_1)) \times \mathcal{S}(p(\mathbf{z}_2)) \times \cdots \times \mathcal{S}(p(\mathbf{z}_K)) \stackrel{def}{=} \mathcal{S}^\times(p(\mathbf{z}))$, where $\times$ denotes Cartesian product.*

Note the factorized support implies that for any composition of $\mathbf{z}_i$ independently encoded from multiple images, there must exist some real image $\mathbf{x}$ represented by $\mathbf{z}$, aligning with our formulation. To achieve the disentangled representation, following the definition, we design mixing strategies that achieve $q_\theta(\mathbf{z}|\mathbf{x})$ that the aggregate distribution $\bar{q}_\theta(\mathbf{z}) = \mathbb{E}_\mathbf{x}[q_\theta(\mathbf{z}|\mathbf{x})]$ has factorized support: $\mathcal{S}(\bar{q}_\theta(\mathbf{z})) = \mathcal{S}^\times(\bar{q}_\theta(\mathbf{z}))$. While the factorized support in the definition implies independent sampling of $\mathbf{z}^i$, we theoretically and empirically show that mixing between two images and $K$ images is equivalent (see Appendix A.2). We now illustrate how we derive a mixing strategy between two images based on the specific compositional structure of factors for attribute and object disentanglement.

**Mixing Strategy for Attribute Disentanglement** In attribute disentanglement, it is typically assumed that each scene is composed of $K$ unique factors. For example, a human face consists of fixed set of features such as eyes, a nose, a mouth, and ears, with each factor being distinctive and included only once. It indicates that our mixing strategy should guarantee mutual exclusiveness in mixing $\mathbf{z}^1, \mathbf{z}^2$ to ensure the resulting $\mathbf{z}^c$ always contains $K$ distinct factors. From definition 1, this condition translates into the mutual exclusiveness on support between latent variables, *i.e.*, $\mathcal{S}(p(\mathbf{z}_i)) \neq \mathcal{S}(p(\mathbf{z}_j)), \forall i \neq j$. Based on this compositional structure for attribute disentanglement, we design a corresponding mixing strategy between two images $\mathbf{x}^1, \mathbf{x}^2$ by randomly selecting each latent $\mathbf{z}_i$ exclusively from either $\mathbf{z}_i^1$ or $\mathbf{z}_i^2$ (see Figure 1 (a) above), *i.e.*, each latent $\mathbf{z}_i$ is drawn from one of the two, but never from both.

Specifically, let $I^S$ be a randomly sampled subset of the index set $I = \{1, \ldots, K\}$. The mixing strategy $\pi_{attr}$ for attribute disentanglement is defined as:

$$\pi_{attr}(\mathbf{z}^1, \mathbf{z}^2) = \{\mathbf{z}_j^1 | j \in I^S\} \cup \{\mathbf{z}_j^2 | j \in I - I^S\} \tag{2}$$

Our mixing strategy (Equation 2) shares similarities with the random permutation trick used in Factor-VAE (Kim & Mnih, 2018). FactorVAE enforces a factorized posterior by randomly mixing individual dimensions of the latent representations across different images. However, it assumes statistical independence in the latent space and minimizes the KL divergence between the factorized posterior (the distribution of the randomly mixed samples) and the aggregated posterior (the distribution of the original samples). While effective for disentangling statistically independent factors of variation, this objective is inherently limited to such attribute factors, making it non-trivial to extend to disentangling

other classes of factors, such as objects. In contrast, our approach embeds the inductive bias directly into the mixing strategy itself, enabling the disentanglement of multiple classes of factors without requiring modifications to the objective function.

**Mixing Strategy for Object Disentanglement**   Object-centric learning often assumes that a scene is composed of a set of objects, where all objects are belong to the same class of factors of variation (Greff et al., 2019). For instance, as all objects belong to same class of factor, replacing an object in image with any object from different images remain realistic. Therefore, each disentangled representation $\mathbf{z}_i$ can encode any object, indicating that all $\mathbf{z}_i$ share the same support set, *i.e.*, $\mathcal{S}(p(\mathbf{z}_i)) = \mathcal{S}(p(\mathbf{z}_j))$ for $i, j \in \{1, \ldots, K\}$. Since all $\mathbf{z}_i$ share the same support, for disentangled representation, it satisfies $\mathcal{S}(p(\mathbf{z})) = \mathcal{S}(p(\mathbf{z}_1)) \times \ldots \times \mathcal{S}(p(\mathbf{z}_K)) = \mathcal{S}(p(\mathbf{z}_{r_1})) \times \ldots \times \mathcal{S}(p(\mathbf{z}_{r_K}))$, where $r_i \in \{1, \ldots, K\}$ in definition 1. It indicates that there must exist $\mathbf{z}$ from some image $\mathbf{x}$ for any arbitrary combinations of object representations without considering mutual exclusiveness as in mixing for attributes. This necessitates a mixing strategy that accommodates arbitrary object combinations, enabling the replacement of any $\mathbf{z}_i$ with any $\mathbf{z}_j$. Accordingly, the mixing strategy for object disentanglement involves randomly sampling $K$ elements from the joint set $\mathbf{z}^1 \bigcup \mathbf{z}^2 \in \mathbb{R}^{2K \times D}$. Unlike the mixing strategy for attributes, this approach permits random exchanges between $\mathbf{z}_i^1$ and $\mathbf{z}_j^2$ between different indices (see Figure 1 above). Specifically, denoting $I^{S_n}$ as a randomly sampled subset of the index set $I = \{1, \ldots, K\}$ with cardinality $n$, i.e., $\{I^S | I^S \subseteq I, |I^S| = n\}$. Then the corresponding mixing strategy $\pi_{obj}$ for object disentanglement is defined as:

$$\pi_{obj}(\mathbf{z}^1, \mathbf{z}^2) = \{\mathbf{z}_j^1 | j \in I^{S_n}\} \cup \{\mathbf{z}_j^2 | j \in I^{S_{K-n}}\}, n \sim U(0, K) \tag{3}$$

### 3.3 Learning Objectives

In this section, we illustrate the overall learning objectives to instantiate our framework. Following the recent approaches, our framework is built upon the auto-encoding framework. Specifically, instead of directly reconstructing the image, we minimize a denoising objective using a diffusion decoder, following state-of-the-art methods (Yang et al., 2023; Jung et al., 2024) for both attribute and object disentanglement, as:

$$\mathcal{L}_{\text{Diff}}(\theta, \phi) = \mathbb{E}_{\epsilon \sim \mathcal{N}(\mathbf{0}, \mathbf{I}), t \sim U(0, 1)} \left[ w(t) \cdot \| D_\phi(\mathbf{x}_t, t, E_\theta(\mathbf{x})) - \epsilon \|^2 \right], \tag{4}$$

where $\mathbf{x}_t = \sqrt{\bar{\alpha}_t} \mathbf{x} + \sqrt{1 - \bar{\alpha}_t}$ is a noised image of $\mathbf{x}$ with timestep $t$, $\bar{\alpha}_t = \prod_i^t (1 - \beta_i)$ is a schedule function, and $w(t)$ is the weighting parameter. As we use diffusion decoder $D_\phi$, we use iterative decoding when generating composite image $\mathbf{x}^c$ from the diffusion decoder but omit the expression for notational simplicity. In addition to the auto-encoding objective, we employ two additional objectives: likelihood maximization objective and compositional consistency objective.

**Maximizing Likelihood of Composite Images**   Given $\mathbf{x}^c$ composed by our mixing strategy, we maximize the likelihood of $\mathbf{x}^c$. To maximize the likelihood of the composite image $\mathbf{x}^c$, we leverage a pre-trained diffusion model $G_\psi$ for its reliable likelihood estimations and robust generative performance. Since the denoising loss in diffusion models serves as an upper bound for the negative log-likelihood (Ho et al., 2020), minimizing the denoising loss with respect to $\mathbf{x}^c$ effectively increases the likelihood $p(\mathbf{x}^c)$. However, due to the expensive and noisy computation of gradients in back-propagating through a large diffusion model, we follow (Poole et al., 2022; Jung et al., 2024) and apply an approximated gradient to optimize $p(\mathbf{x}^c)$:

$$\nabla_\theta \mathcal{L}_{\text{Prior}}(\theta) = \mathbb{E}_{t, \epsilon}[w(t)(G_\psi(\mathbf{x}_t^c, t) - \epsilon) \frac{\partial \mathbf{x}^c}{\partial \theta}], \tag{5}$$

where $t \sim \mathcal{U}(t_{\min}, t_{\max})$ is a timestep, $w(t)$ is a timestep-dependent function, $\epsilon \sim \mathcal{N}(\mathbf{0}, \mathbf{I})$ is a Gaussian noise. $\mathbf{x}_t^c = \sqrt{\bar{\alpha}_t} \mathbf{x}^c + \sigma_t \epsilon$ denotes a noised image of $\mathbf{x}^c$ with the forward diffusion process and $w(t)$ is usually set to $\sigma_t^2$ following (Poole et al., 2022).

While Jung et al. (2024) also maximize the compositionality of object-representations with the generative prior, the authors propose reusing $D_\phi$—the diffusion decoder jointly trained with the encoder in Equation 4—for $G_\psi$, optimizing $\nabla_\theta \mathcal{L}'_{\text{Prior}}(\theta) = \mathbb{E}_{t, \epsilon}[w(t)(D_\phi(\mathbf{x}_t^c, t, \mathbf{z}^c) - \epsilon) \frac{\partial \mathbf{x}^c}{\partial \theta}]$ instead of Equation 5. We argue that diffusion decoder $D_\phi$ is in fact estimates $p(\mathbf{x}^c | \mathbf{z}^c)$ rather than $p(\mathbf{x}^c)$, making them unsuitable for estimating $p(\mathbf{x}^c)$. Thus, we instead opt for a separately pre-trained unconditional diffusion model for $G_\psi$.

**Compositional Consistency Loss**   In addition to maximizing the likelihood $p(\mathbf{x}^c)$, we encourage computational consistency between $\mathbf{z}^c$ and $\hat{\mathbf{z}}^c = E_\theta(D_\phi(\mathbf{z}^c))$ to avoid generating realistic images regardless of the given $\mathbf{z}^c$. A straightforward way to promote compositional consistency is to minimize the cosine distance between $\mathbf{z}^c$ and the inverted latent $\hat{\mathbf{z}}^c = E_\theta(D_\phi(\mathbf{z}^c))$. However, our empirical observations reveal that this direct minimization alone is insufficient to prevent misalignment between $\mathbf{x}^c$ and $\mathbf{z}^c$. In practice, we find that $\mathbf{z}$ from all of the images tend to cluster closely together in the latent space, when directly minimizing cosine distance between $\mathbf{z}^c$ and $\hat{\mathbf{z}}^c$. This clustering causes the distance between $\hat{\mathbf{z}}^c$ and $\mathbf{z}^c$ to remain small, even when the generated composite image $\mathbf{x}^c$ does not faithfully correspond to $\mathbf{z}^c$, thereby reducing the effectiveness of the compositional consistency loss. To address this issue, we instead minimize the *relative* distance between $\mathbf{z}^c$ and $\hat{\mathbf{z}}^c$, *i.e.*, the distance relative to negative samples, which are latents from other random images. This prevents the encoder from collapsing the posterior into a single mode, as $\mathbf{z}^c$ must not only match $\hat{\mathbf{z}}^c$ but also be distinguished from negative samples. Formally, we define the compositional consistency loss as:

$$\mathcal{L}_{\text{Con}}(\theta) = -\log \frac{\exp(d(\hat{\mathbf{z}}^c, \mathbf{z}^c)/\tau)}{\sum_{i \in \{1, \ldots, B\}} \exp(d(\hat{\mathbf{z}}^c, \mathbf{z}^i)/\tau)}, \tag{6}$$

where $\tau$ and $d(\cdot)$ denote temperature and cosine similarity, respectively, and $B$ is a batch size. Note that we should consider the correspondence between $\mathbf{z}^c = \{\mathbf{z}_1^c, \ldots, \mathbf{z}_K^c\}$ and $\hat{\mathbf{z}}^c = \{\hat{\mathbf{z}}_1^c, \ldots, \hat{\mathbf{z}}_K^c\}$ to compute the cosine distance. This can be problematic for object disentanglement, as object-disentangled representations can have permuted orders due to our mixing strategy. In this case, we first apply the Sinkhorn-Knopp algorithm (Cuturi, 2013) to compute a soft assignment between $\mathbf{z}^c$ and $\hat{\mathbf{z}}^c$, then use the assignment-weighted sum of the distances to compute the loss.

**Overall Objectives**   In summary, our framework is built upon an auto-encoding framework, which is implemented with denoising objective. To maximize the compositionality of composite images, we maximize the likelihood of the composite image $\mathbf{x}^c$ with pre-trained diffusion model $G_\psi$, and enforce compositional consistency to ensure resulting $\mathbf{x}^c$ consistent to $\mathbf{z}^c$. The overall objective is given as:

$$\mathcal{L}_{\text{Total}}(\theta, \phi) = \mathcal{L}_{\text{Diff}}(\theta, \phi) + \lambda_{\text{Prior}}\mathcal{L}_{\text{Prior}}(\theta) + \lambda_{\text{Con}}\mathcal{L}_{\text{Con}}(\theta), \tag{7}$$

where $\lambda_{\text{Prior}}$ and $\lambda_{\text{Con}}$ controls the relative importance of the objectives. Note that these objectives are not tailored specific to each factor of variation, but instead shared for both attribute and object disentanglement.

## 4 EXPERIMENT

**Implementation Details**   We use the same encoder and decoder architectures as the baselines (Yang et al., 2023; Jung et al., 2024) for a fair comparison. Following the state-of-the-art methods in attribute (Yang et al., 2023) and object (Jiang et al., 2023) disentanglements, we employ a pre-trained VAE (Rombach et al., 2022) to represent an image as a latent feature and a latent diffusion model (Rombach et al., 2022) for the decoder $D_\phi$. Since the diffusion decoder operates on VAE features, we design image encoder to take VAE features as an input. When generating the image $\mathbf{x}^c$ from $\mathbf{z}^c$, we iteratively decode images using only a few steps (1 to 4 steps) following DDIM (Song et al., 2020) to efficiently reduce the costly iterative decoding process. As back-propagating the gradients through all of the denoising step is often computationally prohibitive, we follow recent work in diffusion-based optimization (Clark et al., 2023; Prabhudesai et al., 2023) and truncate the gradient at the last iteration of decoding. Also, to ensure reliable image generation via few-step decoding, we use a v-prediction objective when training the diffusion model (Salimans & Ho, 2022). For the generative prior $G_\psi$, we train an unconditional diffusion model on each training dataset from scratch. More implementation details can be found in the Appendix A.4.

### 4.1 ATTRIBUTE DISENTANGLEMENT

**Datasets**   We evaluate our method on three standard datasets in disentangled representation learning. Shapes3D (Kim & Mnih, 2018) consists of 3D shapes with 6 ground truth factors. Cars3D (Reed et al., 2015) is a dataset of 3D car models with 3 ground truth factors. MPI3D (Gondal et al., 2019) contains physical 3D objects with 7 factors of variation. All experiments are conducted at a 64x64 image resolution, following (Ren et al., 2022; Yang et al., 2023).

Table 1: Comparisons of attribute disentanglement on the FactorVAE score and DCI disentanglement metrics. Our method achieves state-of-the-art performance in almost all of the datasets, except FactorVAE score in Cars3D.

| Method | Cars3D | | Shapes3D | | MPI3D | |
|---|---|---|---|---|---|---|
| | FactorVAE | DCI | FactorVAE | DCI | FactorVAE | DCI |
| FactorVAE (Kim & Mnih, 2018) | 0.906±0.052 | 0.161±0.019 | 0.840±0.066 | 0.611±0.082 | 0.152±0.025 | 0.240±0.051 |
| $\beta$-TCVAE (Chen et al., 2018) | 0.855±0.082 | 0.140±0.019 | 0.873±0.074 | 0.613±0.114 | 0.179±0.017 | 0.237±0.056 |
| InfoGAN-CR (Lin et al., 2020b) | 0.411±0.013 | 0.020±0.011 | 0.587±0.058 | 0.478±0.055 | 0.439±0.061 | 0.241±0.056 |
| LD (Voynov & Babenko, 2020) | 0.852±0.039 | 0.216±0.072 | 0.805±0.064 | 0.380±0.064 | 0.391±0.039 | 0.196±0.038 |
| GS (Härkönen et al., 2020) | 0.932±0.018 | 0.209±0.031 | 0.788±0.091 | 0.284±0.034 | 0.464±0.036 | 0.229±0.042 |
| DisCo (Ren et al., 2022) | 0.855±0.074 | 0.271±0.037 | 0.877±0.031 | 0.708±0.048 | 0.371±0.030 | 0.292±0.024 |
| DisDiff-VQ (Yang et al., 2023) | **0.976±0.018** | 0.232±0.019 | 0.902±0.043 | 0.723±0.013 | 0.617±0.070 | 0.337±0.057 |
| **Ours** | 0.877±0.089 | **0.365±0.073** | **0.975±0.059** | **0.837±0.105** | **0.668±0.055** | **0.409±0.035** |

Figure 2: Qualitative results on Shapes3D and Cars3D. We swap each latent of source image with the one in target image. Our model successfully identifies six underlying factors in shape3D. In Cars3D, our method discovered three factors including appearance, direction, axis.

**Evaluation Metrics** We use two evaluation metrics: the FactorVAE (Kim & Mnih, 2018) score and the DCI (Eastwood & Williams, 2018) metric. The FactorVAE score measures disentanglement using majority vote classifiers trained to predict the changing ground-truth factor. The DCI metric quantifies disentanglement by assessing each dimension's dominance in predicting each attribute. Since our method induces a vector-wise disentanglement, we perform PCA as post-processing on the representation before evaluation, following (Du et al., 2021; Yang et al., 2023).

**Baselines** We compare our method with state-of-the-art baselines: (1) VAE-based methods, including FactorVAE Kim & Mnih (2018) and $\beta$-TCVAE Chen et al. (2018), (2) GAN-based methods, including InfoGAN-CR Lin et al. (2020b), GANspace (GS) Härkönen et al. (2020), LatentDiscovery (LD) Voynov & Babenko (2020), and DisCo Ren et al. (2022), and (3) the diffusion-based model DisDiff Yang et al. (2023). We mostly follow the experimental settings in DisDiff and use the same encoder and diffusion decoder architecture as DisDiff.

**Main Results** We first report the comparison results of our method with baselines for attribute disentanglement in Table 1. Our method outperforms all baselines on the Shapes3D and MPI3D datasets by a clear margin, achieving 8% higher FactorVAE scores and 15.7% to 21.4% higher DCI metrics compared to the second-best baselines. For the Cars3D dataset, our method achieves the best DCI metric. Notably, on Shapes3D and MPI3D datasets, our method outperforms the state-of-the-art baseline DisDiff (Yang et al., 2023) with substantial margin. This indicates the effectiveness of our objective in directly enforcing the support factorization between latent representations via our mixing strategy for disentangling factors, compared to using an approximate measure such as the upper bound of mutual information between latents.

Note that our method also significantly outperforms FactorVAE (Kim & Mnih, 2018), which similarly utilizes random mixing of representations. We hypothesize that our method benefits from flexible

choice of model architectures. Specifically, FactorVAE is specifically designed to disentangle between latent dimensions within VAE framework to explicitly minimize Total Correlation. In contrast, our framework can freely choose the model architecture, so our model benefits from vector-wise disentanglement and expressive decoder, *i.e.*, diffusion model, which are known to have better disentanglement and representation quality. Overall, the quantitative results demonstrate the effectiveness of our model in attribute disentanglement.

To further analyze the quality of our disentangled representations, we perform image generation by swapping the latent representations between images in Fig. 2. We first encode a randomly sampled target image and six randomly sampled source images into $K$ latent representations each. For each $k \in \{1, ..., K\}$, we then construct swapped representations by replacing the $k$ th latent representation from the target image with the $k$ th latent representation from the source images and decode these swapped representations. The results demonstrate the effectiveness of our method in attribute disentanglement and compositional image generation. Surprisingly, in the Shapes3D dataset, our method successfully identifies all six ground-truth factors of variation. In the Cars3D dataset, our method captures three independent factors, enabling controlled manipulation of each factor.

## 4.2 Object Disentanglement

**Datasets**    We evaluate our method for object disentanglement on three multi-object datasets. CLEVR-Easy (Singh et al., 2022b) contains images with 2-3 objects in different colors, shapes, and positions. CLEVR (Johnson et al., 2017) consists of images containing 3-10 objects, further differing in size and material compared to CLEVR-Easy. In CLEVR-Tex (Singh et al., 2022b), textures are added to objects and backgrounds of the CLEVR dataset, leading more complex scenes with diverse materials. All images in the datasets are center-cropped and resized to $128 \times 128$ pixels.

**Evaluation Protocol**    We evaluate the quality of object representations through an object property prediction task, following (Jiang et al., 2023; Jung et al., 2024). For each property, we train a network to predict the property based on frozen object representations. Correspondences between the each representation and GT objects are determined through Hungarian matching using masks. For baselines, slot-attention masks are used for matching. In contrast, as our method does not produce masks, we identify the corresponding region of the object representation by averaging the differences in output images when we compose each representation with other representations. For the classifier, we employ a 2-layer MLP with a hidden dimension of 256. We report accuracy for categorical properties and mean squared error (MSE) for continuous properties.

**Baselines**    We compare our method with object-centric learning methods leveraging slot-attention: SA (Locatello et al., 2020) and SLASH (Kim et al., 2023). Also, we compare our method against state-of-the-art methods using the diffusion decoders: LSD (Jiang et al., 2023) and L2C (Jung et al., 2024). It's worth noting that ours does not employ slot attention or any kinds of spatial-exclusiveness biases. For a fair comparison, we employ the same encoder architecture across all baselines including ours, and all diffusion-based methods share the same decoder.

**Main Results**    Tab. 2 presents the results of the object property prediction task. Our method achieves competitive performance compared to state-of-the-art baselines, LSD (Jiang et al., 2023) and L2C (Jung et al., 2024), demonstrating its effectiveness in object-centric learning. Notably, our method outperforms LSD on CLEVR-Tex and achieves comparable performance on CLEVR and CLEVR-Easy. Considering the primary difference between LSD and our method is the use of slot attention versus the compositionality maximization by mixing strategy, our method's competitive performance validates the effectiveness of our mixing strategy as a strong inductive bias for object disentanglement. In comparison to L2C, our method achieves better performance on CLEVR and CLEVR-Easy while being competitive on CLEVR-Tex. In CLEVR dataset, we observed that slot attention in L2C got undesirable positional biases. Since L2C maximizes conditional likelihood $p(\mathbf{x}^c|\mathbf{z}^c)$, it can be achieved by local encoding and decoding instead of maximizing $p(\mathbf{x})$. Overall, the competitive performance of our method compared to strong baselines verifies that our mixing strategy provides robust inductive bias for object-centric learning.

We further explore the compositionality of our latent representations in Fig. 3. Given pairs of images, we encode each image into $K$ object representations and construct a mixed representation by

Table 2: Comparison of object disentanglement on property prediction. For the position* property of CLEVREasy dataset, we use the discrete labels provided in the dataset and reports the accuracy.

| Method | CLEVREasy | | | CLEVR | | | | CLEVRTex | | |
|---|---|---|---|---|---|---|---|---|---|---|
| | Shape (↑) | Color (↑) | Position* (↑) | Shape (↑) | Color (↑) | Material (↑) | Position (↓) | Shape (↑) | Material (↑) | Position (↓) |
| SA | 72.25 | 72.33 | 44.08 | 79.4 | 91.30 | 93.18 | 0.064 | 30.44 | 7.890 | 0.482 |
| SLASH | 86.06 | 89.23 | 46.97 | 83.28 | 92.26 | 93.16 | 0.078 | 53.13 | 37.49 | 0.148 |
| LSD | 96.03 | 98.05 | 50.29 | 87.66 | 91.46 | 94.87 | 0.062 | 68.25 | 51.54 | 0.197 |
| L2C | 92.78 | 93.57 | 47.62 | 73.61 | 74.03 | 86.93 | 0.168 | 71.54 | 51.62 | 0.116 |
| **Ours** | 95.81 | 95.38 | **50.72** | 87.04 | **93.93** | 94.81 | **0.032** | 70.90 | **52.2** | 0.133 |

Figure 3: Qualitative results on object-wise manipulation in CLEVR and CLEVRTex. Objects depicted with red arrows are replaced by the the one depicted with green arrows. Successful object-wise manipulation verifies that our method successfully disentangles the objects. We also find *empty* latent (depicted with $\phi$), which makes our approach capable of handling varying number of objects.

randomly exchanging one latent between images. The mixed representations are then decoded with the decoder to produce final composite images. In Fig. 3, we replace one object (depicted with red arrow) from first column with the object (depicted with red arrow) from first row's image. In second to fifth column, we identify successful insertion of the individual objects depicted in to first row into the first column's image. Meanwhile, the objects depicted with red arrows are successfully removed from the original scene. It demonstrates that our method successfully disentangle individual objects. Notably, in fifth row and fifth column, we observe that our method allows the emergence of latent encoding empty information. When manipulate such latent, it does not add any of the objects (in fifth column) or remove none of the objects (in fifth row) from the original images. It highlights that our method is capable of capturing varying number of objects.

## 4.3 ABLATION STUDY

**Impact of Losses** We conduct an ablation study on the impact of each term in our objectives and present the results in Tab. 3. The results indicate that incorporating all three losses of diffusion ($\mathcal{L}_{\text{Diff}}$), prior ($\mathcal{L}_{\text{Prior}}$), and cycle loss ($\mathcal{L}_{\text{Con}}$) is essential for our method. In attribute disentanglement learning, sequentially adding each loss term consistently improves performance, with the best results achieved when all losses are combined. In contrast, for object disentanglement learning, clear performance gains across all three property predictions are observed only when using all loss terms together, possibly due to differences in the mixing strategy.

**Impact of Mixing Strategy** We investigate the importance of an appropriate mixing strategy for attribute and object disentanglement learning. We experimented with object mixing and attribute mixing applied interchangeably to attribute disentanglement learning and object disentanglement learning, respectively. The results are shown in the bottom three rows of Tab. 3. The results show that the interchanged mixing strategy significantly degrades performance, in both attribute and object disentanglement learning, highlighting the importance of a proper mixing strategy in our method.

Table 3: Ablation study on our method. We investigate the impact of each learning objective and mixing strategy. It confirms that our method work best with all of the objectives and proper choice of mixing strategy improves disentanglement.

| | | Shape3D | | | Clevr | |
|---|---|---|---|---|---|---|
| | | FactorVAE | DCI | Shape ($\uparrow$) | Color ($\uparrow$) | Position ($\downarrow$) |
| Impact of Losses | $\mathcal{L}_{\text{Diff}}$ | 0.492 | 0.175 | 62.270 | 88.580 | 0.111 |
| | $\mathcal{L}_{\text{Diff}} + \mathcal{L}_{\text{Prior}}$ | 0.597 | 0.224 | 63.393 | 86.943 | 0.126 |
| | $\mathcal{L}_{\text{Diff}} + \mathcal{L}_{\text{Con}}$ | 0.769 | 0.597 | 64.210 | 80.279 | 0.116 |
| | $\mathcal{L}_{\text{Diff}} + \mathcal{L}_{\text{Prior}} + \mathcal{L}_{\text{Con}}$ | **1.000** | **0.887** | **87.039** | **93.928** | **0.032** |
| Impact of Mixing Strategy | Attribute mixing | 1.000 | 0.887 | 65.236 | 80.520 | 0.119 |
| | Object mixing | 0.634 | 0.127 | **87.040** | **93.928** | **0.033** |

(a) OOD example 1    (b) OOD example 2    (c) Decoded images from different mixing strategy

Figure 4: Qualitative analysis on our method. Our analysis verifies that our method can generalize to out-of-distribution (OOD) scenes (a), (b) and highlights the importance of choosing an appropriate mixing strategy (c).

**More qualitative analysis**    In Fig 4-(a, b), we observe that our method is capable of generating out-of-distribution (OOD) examples that do not exist in the dataset, but can be created through composition. Notably, using the CLEVR-Easy dataset, which comprises images with 2-3 objects, our method can generate high-quality images containing either a single object or 4 objects through composition, by inserting or removing the representation that does not encodes object. In Fig 4-(c), we compare images composed from models trained using different mixing strategies: object mixing and attribute mixing. As demonstrated in the main results and our ablation study, the object mixing strategy allows for object-level manipulation. In contrast, while the attributed mixing strategy, also supported by the prior loss, produces images of reasonable quality, but it does not achieve object-level modifications. Specifically, when object slots are swapped, the changes in the image are not confined to a single object but also alter the properties of other objects.

## 5   LIMITATIONS AND FUTURE WORK

While our method aims to identify underlying factors of variations by compositionality within the representation, discovered factor may not exactly aligned to ground-truth factors. As data may not be decomposed in a unique way, it's challenging to discover the exact decomposition of data using our method. In this work, our framework demonstrates how to uncover the general factors of variation, focusing on the representative examples in the field (*e.g.*, attributes and objects.). For future work, we will further explore nuanced and intricate factors of variation within the data.

## 6   CONCLUSION

In this paper, we introduced a unified framework for disentangled representation learning that is compatible with both attribute and object disentanglement. We formulate disentangled representation learning as the process of maximizing compositionality within the representation, enabling both attribute and object disentanglement by controlling only the composition operator. Although compatible with both attribute and object disentanglement, our method achieved competitive performance against strong baselines in each domain.

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

# A  APPENDIX

## A.1  RELATED WORK

**Disentangled Representation Learning**    Disentangled representation learning for attribute disentanglement heavily rely on regularization terms in learning objectives (Burgess et al., 2017; Chen et al., 2016; Kim & Mnih, 2018; Chen et al., 2018; Ren et al., 2022; Yang et al., 2023). VAE-based models (Burgess et al., 2017; Kim & Mnih, 2018; Chen et al., 2018) demonstrates that controlling the importance of total correlation between latent dimensions hidden in the ELBO bounds encourages to disentangle independent factors. Empowered with enhanced generative models such as GANs (Goodfellow et al., 2020) and diffusion models (Ho et al., 2020), (Chen et al., 2016; Lin et al., 2020b) optimizes the mutual information between latents and generated images by GANs, and (Ren et al., 2022; Yang et al., 2023) proposed to optimize contrastive loss (Oord et al., 2018) or mutual information between the latents using pretrained GANs and diffusion model, respectively. Such information-theoretic approaches have shown promising disentangling capabilities, but it becomes challenging when a scene does not consist of fixed combination of factors, especially when there exists repeated appearances of the same factors, as seen in object-centric scenes.

**Object-Centric Learning**    Built on the observation that each pixel in a scene must correspond exclusively to an unique object, the spatial-exclusive mechanism has been recognized as a key inductive bias in object-centric learning (Burgess et al., 2019; Greff et al., 2019; Engelcke et al., 2020; Locatello et al., 2020; Kim et al., 2023; Singh et al., 2022a). Early attempts in object-centric learning employed spatial masks to compose independently decoded RGB images from each latent (Burgess et al., 2019; Greff et al., 2019; Lin et al., 2020a; Engelcke et al., 2020). In addition to the spatial-exclusive bias, iterative refinement of each latent representation gradually improves the initially inaccurate spatial association between each latent and the pixels of the image (Greff et al., 2019). In slot attention (Locatello et al., 2020), each latent (slot), is randomly initialized and iteratively refined by a dot-product attention mechanism normalized over the slots. This mechanism induces competition between the slots to bind to spatial locations in the scene. Empowered by strong generative models combined with Slot Attention, recent studies (Singh et al., 2022a; Jiang et al., 2023; Wu et al., 2023; Jung et al., 2024) have demonstrated remarkable performance in unsupervised object discovery on complex real-world datasets. While these architectural biases excel at object disentanglement, strong assumption on spatial exclusiveness limits their applicability to disentangling non-spatial exclusive factors, such as attributes.

## A.2  PROOF ON EQUIVALENCE BETWEEN MIXING TWO AND MULTIPLE IMAGES.

In this section, we explain why the random mixing between two images (*i.e.*, $\mathbf{z}^c = \pi(\mathbf{z}^1, \mathbf{z}^2)$) can replace the random composition of $\mathbf{z}_i$ from $K$ images. Formally, we will show that:

$$If \, \mathcal{S}(p(\mathbf{z})) = \mathcal{S}(p(\mathbf{z}^c)) \quad \text{then} \quad \mathcal{S}(p(\mathbf{z})) = \mathcal{S}^{\times}(p(\mathbf{z})), \tag{8}$$

where the factorized support $\mathcal{S}^{\times}(p(\mathbf{z})) = \mathcal{S}(p(\mathbf{z}_1)) \times \mathcal{S}(p(\mathbf{z}_2)) \times \cdots \times \mathcal{S}(p(\mathbf{z}_K))$ represents the random composition of each latent variable $\mathbf{z}_i$ from $K$ images.

*Proof.* Given $\mathcal{S}(p(\mathbf{z})) = \mathcal{S}(p(\mathbf{z}^c))$, we can prove the followings:

1. If $p(\mathbf{z}_1)p(\mathbf{z}_2) > 0$ then $p(\mathbf{z}_1, \mathbf{z}_2) > 0$.
   Note that $p(\mathbf{z}_1) > 0$ and $p(\mathbf{z}_2) > 0$ ($\Leftrightarrow p(\mathbf{z}_1)p(\mathbf{z}_2) > 0$) indicates the existence of $\mathbf{z}^1, \mathbf{z}^2$ with $\mathbf{z}_1^1 = \mathbf{z}_1, \mathbf{z}_2^2 = \mathbf{z}_2$. By mixing $\mathbf{z}^1$ and $\mathbf{z}^2$, we can compose $\mathbf{z}^*$ where $\mathbf{z}_1^* = \mathbf{z}_1, \mathbf{z}_2^* = \mathbf{z}_2$. Then, by the definition of the support that $\mathcal{S}(p(\mathbf{z})) = \{\mathbf{z}|p(\mathbf{z}) > 0\}$ and the given condition $\mathbf{z}^* \in \mathcal{S}(p(\mathbf{z}^c)) = \mathcal{S}(p(\mathbf{z})), p(\mathbf{z}_1, \mathbf{z}_2) \geq p(\mathbf{z}^*) > 0$.

2. Assume that for some $k \geq 2$, if $\prod_{i=1}^{k} p(\mathbf{z}_i) > 0 \rightarrow p(\mathbf{z}_1, \mathbf{z}_2, \ldots, \mathbf{z}_k) > 0$ then $\prod_{i=1}^{k+1} p(\mathbf{z}_i) > 0 \rightarrow p(\mathbf{z}_1, \mathbf{z}_2, \ldots, \mathbf{z}_k, \mathbf{z}_{k+1}) > 0$.
   Note that $\prod_{i=1}^{k+1} p(\mathbf{z}_i) > 0$ implies $p(\mathbf{z}_{k+1}) > 0$ and $\prod_{i=1}^{k} p(\mathbf{z}_i) > 0$. By the given assumption, $p(\mathbf{z}_1, \mathbf{z}_2, \ldots, \mathbf{z}_k) > 0$ and there exists $\mathbf{z}^1, \mathbf{z}^2$ where $\mathbf{z}_i^1 = \mathbf{z}_i$ for $i \in \{1, \ldots, k\}$ and $\mathbf{z}_{k+1}^2 = \mathbf{z}_{k+1}$. By mixing $\mathbf{z}^1$ and $\mathbf{z}^2$, we can compose $\mathbf{z}^*$ where $\mathbf{z}_i^* = \mathbf{z}_i$ for $i \in \{1, \ldots, k+1\}$. As a result, by the given condition $\mathbf{z}^* \in \mathcal{S}(p(\mathbf{z}^c)) = \mathcal{S}(p(\mathbf{z}))$, $p(\mathbf{z}_1, \mathbf{z}_2, \ldots, \mathbf{z}_k, \mathbf{z}_{k+1}) \geq p(\mathbf{z}^*) > 0$.

| # of samples for mixing | Factor VAE | DCI |
|:---:|:---:|:---:|
| 2 | 0.975±0.040 | 0.837±0.105 |
| 64 | 0.966±0.032 | 0.802±0.088 |

Table 4: Effects of number of samples used in mixing strategy

3. By mathematical induction, we conclude that if $\prod_{i=1}^{K} p(\mathbf{z}_i) > 0$ then $p(\mathbf{z}) > 0$.

Note that (3) implies $\mathcal{S}(p(\mathbf{z})) = \mathcal{S}^{\times}(p(\mathbf{z}))$, since $\mathcal{S}^{\times}(p(\mathbf{z}))$ can be expressed as $\{\mathbf{z}|p(\mathbf{z}_i) > 0\}$. By using mathematical induction, we have proved that random mixing between two images can replace the random composition of multiple images to achieve disentanglement.

### A.3 EMPIRICAL RESULTS DIFFERENCE BETWEEN MIXING TWO AND MULTIPLE IMAGES.

In additional to theoretic result, we provide empiricial results on our mixing strategy between two and multiple images (we use 64 here) are equivalent. We conduct experiments on attribute disentanglement with three different seeds and report FactorVAE and DCI in Table 4. We identified there is no meaningful difference between mixing two or 64 images, which supports our theoretical result.

### A.4 ADDITIONAL IMPLEMENTATION DETAILS

In this section, we provide additional implementation details. When we train our method, we fix batch size of 64 and learning rate of 0.0001 across all of the experiments. We use $\lambda_{Prior} = 1$ and $\lambda_{Con} = 0.01$ for all experiments except $\lambda_{Con} = 0.1$ for the experiments in MPI3D dataset. We fix number of latents $K = 10$ in attribute disentanglement experiment following the best configuration of DisDiff (Yang et al., 2023) and $K = 4, 11, 11$ for CLEVREasy, CLEVR, CLEVRTex, respectively, for object disentanglement.

Table 12,7,8,14 summarizes the hyper-parameters of our encoder and decoder architectures used in the experiments. Following DisDiff (Yang et al., 2023) and LSD (Jiang et al., 2023), we employ pretrained vq-vae [1] and kl-regularized auto-encoder model [2] in attribute distentanglement and object disentanglement, respectively. In attribute disentanglement experiment, the encoder maps the input $\mathbf{x}$ into 1-dimensional vector $\mathbf{z} \in \mathbb{R}^{KD}$ and we uniformly divide it into $K$ latents. In object disentanglement experiment, to support the mapping from varying number of inputs (*e.g.*, different spatial resolutions of UNet feature) into $K$ latent representations, we adopt QFormer (Li et al., 2023). Specifically, we have $K$ learnable queries $\{\mathbf{q}\}^{K} \in \mathbb{R}^{K \times D}$ and those queries are updated via multiple self attention layers and cross attention layers, where the keys and values are linearly projected from unet feature of $\mathbf{x}$. For QFormer, we use 4 layers with 8 attention heads and hidden dimension of 256.

### A.5 MATCHING TECHNIQUE

We have developed a technique to identify the specific region corresponding to an object's representation based on composed images of that representation. For a given target object representation, we first random sample multiple images and encode them into object representations. For each image, we then replace one object representation with the target object representation and decode the mixed representations. The images generated from this composed representation may include the target object if it is appropriately encoded. To determine the object region, we measure the RGB variance between the generated images. Additionally, we use the original image containing the target object representation and select the region that closely matches the original. Finally, we combine two metrics—the variance and the distance to the original image—to accurately specify the region.

---

[1]     https://huggingface.co/stabilityai/sd-vae-ft-ema-original
[2]     https://ommer-lab.com/files/latent-diffusion/celeba.zip

```
Conv 3 × 3 × 3 × 128, stride=1
BatchNorm2d
ReLU
Conv 3 × 3 × 128 × 128, stride=1
BatchNorm2d
ReLU
Conv 3 × 3 × 128 × 128, stride=1
BatchNorm2d
ReLU
Conv 3 × 3 × 128 × 128, stride=1
BatchNorm2d
ReLU
ResBlock 3 × 3 × 128 × 128, stride=1
BatchNorm2d
ReLU
ResBlock 3 × 3 × 128 × 128, stride=1
BatchNorm2d
ReLU
FC 4096 × 10
```

Table 5: Encoder Architecture used in attribute disentanglement.

```
ReLU
Conv 3 × 3 × 128 × 128, stride=1
BatchNorm2d
ReLU
Conv 3 × 3 × 128 × 128, stride=1
```

Table 6: ResBlock in the Encoder

| | |
|---|---|
| Input Resolution | 16 |
| Input Channels | 3 |
| Input Channels | 4 |
| $\beta$ scheduler | Linear |
| Mid Layer Attention | Yes |
| # Res Blocks / Layer | 2 |
| # Heads | 8 |
| Base Channels | 64 |
| Attention Resolution | [1,2,4,4] |
| Channel Multipliers | [1,2,4,4] |

Table 7: Decoder Architecture used in attribute disentanglement

| | |
|---|---|
| Input Resolution | 16 |
| Input Channels | 3 |
| Output Resolution | 16 |
| Self Attention | Middle Layer |
| Base Channels | 128 |
| Channel Multipliers | [1,1,2,4] |
| # Heads | 8 |
| # Res Blocks / Layer | 1 |

Table 8: Unet Encoder Architecture used in object disentanglement.

| | |
|---|---|
| Input Resolution | 16 |
| Input Channels | 4 |
| $\beta$ scheduler | Linear |
| Mid Layer Attention | Yes |
| # Res Blocks / Layer | 2 |
| # Heads | 8 |
| Base Channels | 192 |
| Attention Resolution | [1,2,4,4] |
| Channel Multipliers | [1,2,4,4] |

Table 9: Decoder Architecture used in object disentanglement.

| | |
|---|---|
| Input Resolution | 16 |
| Input Channels | 3 |
| $\beta$ scheduler | Linear |
| Mid Layer Attention | Yes |
| # Res Blocks / Layer | 2 |
| # Heads | 8 |
| Base Channels | 64 |
| Attention Resolution | [1,2,4,4] |
| Channel Multipliers | [1,2,4,4] |

Table 10: Generative Prior Architecture used in attribute disentanglement.

| | |
|---|---|
| Input Resolution | 16 |
| Input Channels | 4 |
| $\beta$ scheduler | Linear |
| Mid Layer Attention | Yes |
| # Res Blocks / Layer | 2 |
| # Heads | 8 |
| Base Channels | 192 |
| Attention Resolution | [1,2,4,4] |
| Channel Multipliers | [1,2,4,4] |

Table 11: Generative Prior Architecture used in object disentanglement.

```
Conv 3 × 3 × 3 × 128, stride=1
BatchNorm2d
ReLU
Conv 3 × 3 × 128 × 128, stride=1
BatchNorm2d
ReLU
Conv 3 × 3 × 128 × 128, stride=1
BatchNorm2d
ReLU
Conv 3 × 3 × 128 × 128, stride=1
BatchNorm2d
ReLU
ResBlock 3 × 3 × 128 × 128, stride=1
BatchNorm2d
ReLU
ResBlock 3 × 3 × 128 × 128, stride=1
BatchNorm2d
ReLU
FC 4096 × 10
```

Table 12: Encoder Architecture used in attribute disentanglement.

```
ReLU
Conv 3 × 3 × 128 × 128, stride=1
BatchNorm2d
ReLU
Conv 3 × 3 × 128 × 128, stride=1
```

Table 13: ResBlock in the Encoder

| | |
|---|---|
| Input Resolution | 16 |
| Input Channels | 4 |
| $\beta$ scheduler | Linear |
| Mid Layer Attention | Yes |
| # Res Blocks / Layer | 2 |
| # Heads | 8 |
| Base Channels | 192 |
| Attention Resolution | [1,2,4,4] |
| Channel Multipliers | [1,2,4,4] |

Table 14: Generative Prior Architecture used in object disentanglement.

## A.6 COMPUTING RESOURCES

We conduct all our experiments on a GPU Server consists of Intel Xeon Gold 6230 CPU, 256GB RAM, and 8 NVIDIA RTX 3090 GPUs (with 24GB VRAM), or 8 NVIDIA RTX 6000 GPUs (with 48GB VRAM). It takes about 24 GPU hours and from 36 to 48 GPU hours for attribute and object disentanglement experiment, respectively.

Table 15: Quantitative Results of unsupervised segmentation in CLEVR dataset

| Method | FG-ARI | | mIoU | | mBO | |
|---|---|---|---|---|---|---|
| | Slot-Attention | SBD Mask | Slot-Attention | SBD Mask | Slot-Attention | SBD Mask |
| LSD | 82.00 | **91.74** | 22.69 | 25.59 | 22.98 | 25.84 |
| L2C | 54.01 | 80.05 | 19.30 | 25.61 | 20.36 | 26.33 |
| Ours | - | 91.20 | - | **26.54** | - | **26.65** |

Table 16: Quantitative Results of unsupervised segmentation in CLEVRTex dataset

| Method | FG-ARI | | mIoU | | mBO | |
|---|---|---|---|---|---|---|
| | Slot-Attention | SBD Mask | Slot-Attention | SBD Mask | Slot-Attention | SBD Mask |
| LSD | 46.54 | 71.64 | 45.87 | 56.26 | 46.93 | 56.75 |
| L2C | 77.07 | 82.55 | 56.59 | 58.33 | 53.25 | 58.68 |
| *Ours* | - | **87.68** | - | **58.88** | - | **59.12** |

## A.7 UNSUPERVISED SEGMENTATION

In this secntion, we additionally measured unsupervised segmentation performance of pretrained encoders. Unlike slot-attention-based methods, our method does not have a built-in mechanism to directly express group memberships between pixels. Therefore, we trained a Spatial Broadcast Decoder (Watters et al., 2019) on top of the frozen latent representations to predict explicit object masks for each latent representation. We train Spatial Broadcast Decoder with a reconstruction loss to recover the original image from frozen latents in an unsupervised manner, and it requires minimal training costs as the encoder remains frozen and the decoder is shallow (See details in Table 17). We trained the decoder for 30k iterations with learning rate of 1e-3. After training Spatial Broadcast Decoder, we extract explicit object masks for each latent and evaluate our method against two strongest baselines in slot-attention-based works, LSD and L2C, on CLEVR and CLEVRTex datasets. For a fair comparison, we evaluate the baselines using both slot-attention mask and object masks obtained by training a Spatial Broadcast Decoder on their frozen slot representations. The results are reported in Table 15, Table 16 and Figure 5.

On the CLEVR dataset, our method achieved the best mIoU and mBO scores, along with comparable FG-ARI. A high FG-ARI indicates that each mask captures complete objects, confirming the effectiveness of our method in object disentanglement. However, we observed that the background is often split across multiple latents. This occurs because the constant backgrounds in CLEVR do not affect compositional generation and therefore avoid penalties from the compositional loss. As constant backgrounds carry null information, this does not impact the quality of object representations. On the CLEVRTex dataset, our method outperformed both LSD and L2C across all three metrics. As shown in Figure 5, our method consistently encodes complete objects into distinct latents, whereas LSD and L2C frequently split objects across multiple latents. This explains the high FG-ARI achieved by our method and verifies its superior object disentanglement. Additionally, unlike CLEVR, as CLEVRTex has various background colors, our model successfully encodes all of the background information into a single latent.

Together, the experiments on unsupervised segmentation also confirm that our method achieves robust object-wise disentanglement. It is notable that our method outperforms baselines in object segmentation without relying on spatial clustering architectures such as slot attention. This highlights the strength of our approach.

Table 17: Spatial Broadcast Decoder Architecture used for Unsupervised Segmentation

| |
| --- |
| ConvTranspose2d $64 \times 64 \times 5 \times 5$, stride=2, padding=2, output_padding=1 |
| ReLU |
| ConvTranspose2d $64 \times 64 \times 5 \times 5$, stride=2, padding=2, output_padding=1 |
| ReLU |
| ConvTranspose2d $64 \times 64 \times 5 \times 5$, stride=2, padding=2, output_padding=1 |
| ReLU |
| ConvTranspose2d $64 \times 64 \times 5 \times 5$, stride=2, padding=2, output_padding=1 |
| ReLU |
| ConvTranspose2d $64 \times 64 \times 5 \times 5$, stride=1, padding=2 |
| ReLU |
| ConvTranspose2d $64 \times 64 \times 5 \times 5$, stride=1, padding=1 |

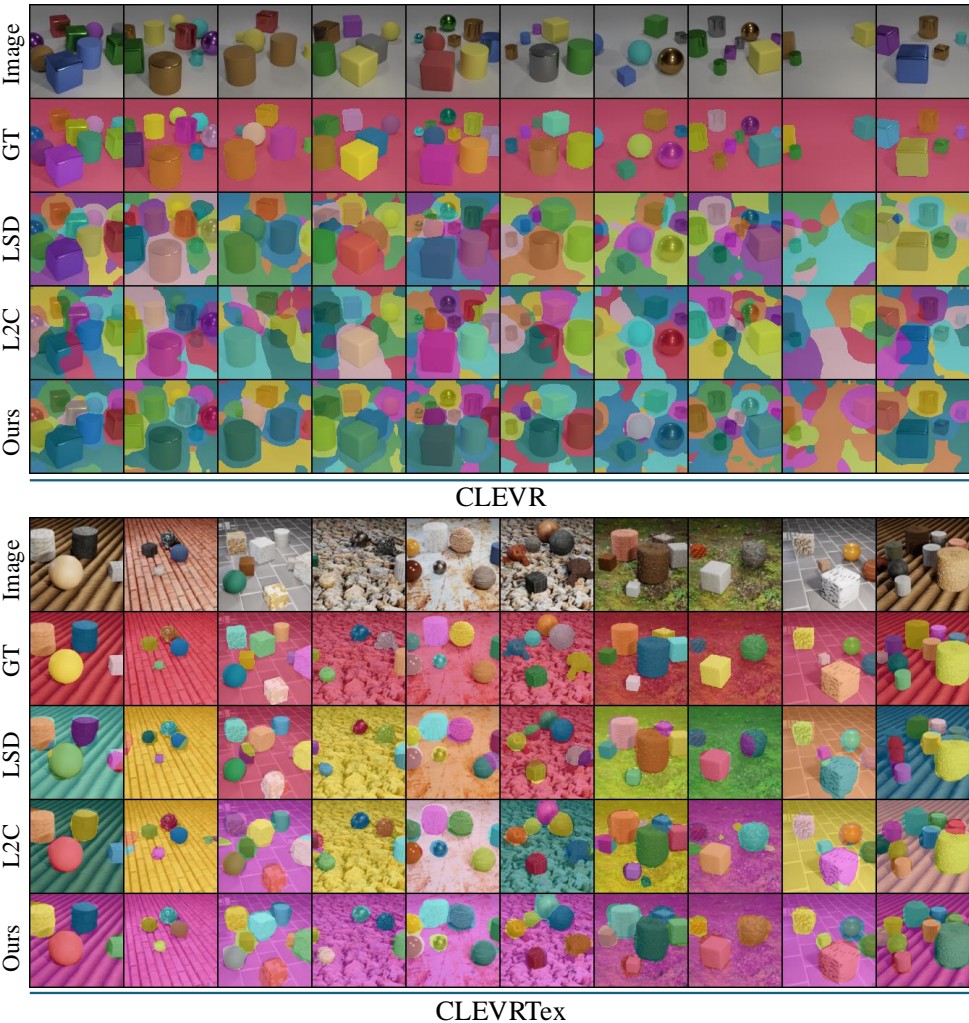

Figure 5: Qualitative results on unsupervised segmentation in CLEVR and CLEVRTex dataset

## A.8 ADDITIONAL EXPERIMENTS ON COMPLEX DATASET

To explore scalability of our method, we additionally conduct experiments on CelebA-HQ for attribute disentanglement and MultiShapeNet (MSN) (Stelzner et al., 2021) for object disentanglement, respectively.

**Attribute Disentanglement in CelebA-HQ**  For the CelebA-HQ dataset, we use the attribute-mixing strategy to disentangle attribute factors. As CelebA-HQ has much more visual complexity compares to synthetic datasets, we replace a shallow encoder used in main experiment with Resnet-18 encoders. For generative prior, we leverage a off-the-shelf unconditional diffusion model [3]. We trained our model for 150k iterations with learning rate of 1e-4.

To verify the disentanglement of the learned representations, we swap each latent vector one by one between two images and present the qualitative results in Figures 6 and Figure 7. In the third columns of each figure, we observe that while the source images lack bangs, the swapped images successfully generate bangs while preserving other attributes. Similarly, in the fourth and fifth columns, the facial expressions and skin tones of the target images are effectively transferred to the source images. These qualitative results demonstrate that our attribute-mixing strategy is capable of disentangling attribute factors, even in complex datasets like CelebA-HQ.

**Object Disentanglement in MultiShapeNet**  We validate our method on MSN dataset with object-wise manipulation and unsupervised segmentation. The model architecture and hyper-parameters were kept the same as in the previous object disentanglement experiments. For the object-wise manipulation task, we encode pairs of images into $N = 5$ object representations and exchange random object latents between the pairs to construct composite images. As shown in Figure 8, our method successfully performed object-level insertion and removal, demonstrating that each latent representation distinctly captures individual objects. This confirms that our approach effectively disentangles object representations within the latent space.

For the unsupervised segmentation task, we measure FG-ARI, mIoU, mBO on object masks following common practices in object-centric literature. As our method does not have a built-in mechanism to directly express group memberships between pixels, we additionally train Spatial Broadcast Decoder (Watters et al., 2019) on the frozen latent representations to predict explicit object masks for each latent representation (please refer to Appendix A.7 for details). The results are reported in Table 18. Among the competitive slot-attention based baselines, our method achieves second-best performances across all of three metrics. The high segmentation scores of L2C are mainly due to its slot-attention-based regularization term (see Equation 8 in the L2C paper), which explicitly encourages the slot masks to align with object shapes. Except for L2C, our method outperforms rests of the baselines (LSD, SLATE) across all metrics, even though ours does not employ any of a spatial clustering mechanism like slot attention. These results demonstrate the effectiveness of our framework in disentangling object representations in a complex dataset.

---

[3]    https://huggingface.co/CompVis/ldm-celebahq-256

Source   Target   Bang   Smile   Skin

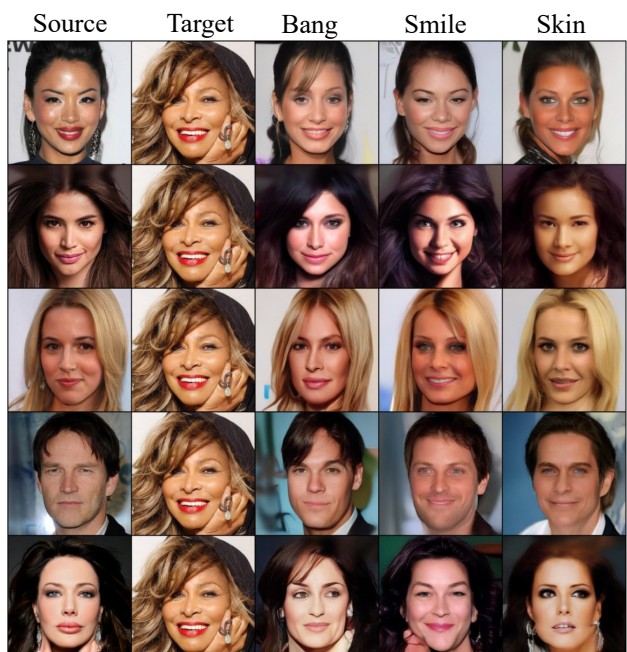

Figure 6: Qualitative results on unsupervised segmentation. We replace source latent representation to target latent representation one by one.

Source   Target   Bang   Smile   Skin

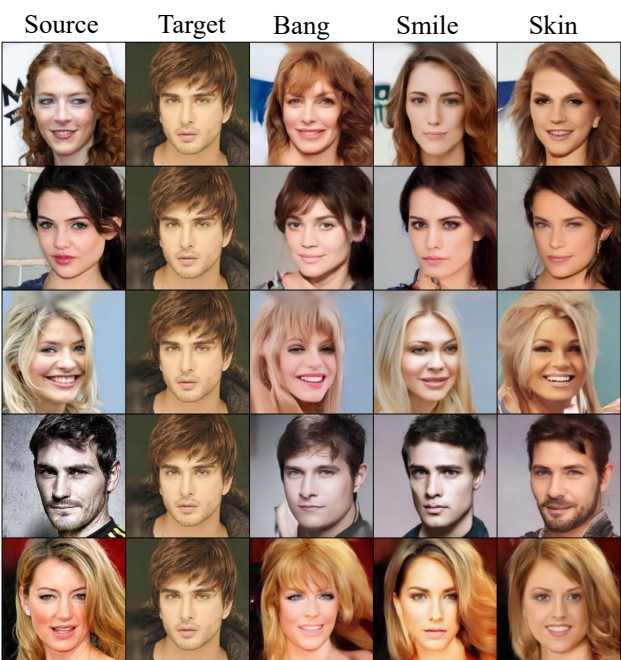

Figure 7: Qualitative results on unsupervised segmentation. We replace source latent representation to target latent representation one by one.

Table 18: Quantitative Results on unsupervised segmentation in MSN dataset. All the values of SLATE+, LSD, L2C are from L2C paper.

| Model | FG-ARI | mIoU | mBO |
|---|---|---|---|
| SLATE+* | 70.44 | 15.55 | 15.64 |
| LSD* | 67.72 | 15.39 | 15.46 |
| L2C* | **89.8** | **59.21** | **59.4** |
| *Ours* | 76.92 | 24.19 | 24.3 |

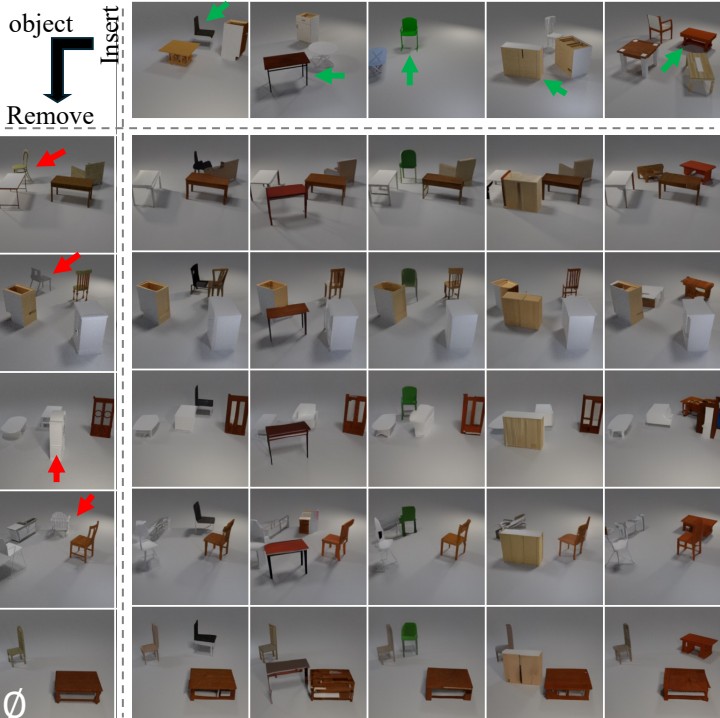

Figure 8: Qualitative results on object-wise manipulation in MSN dataset. Objects depicted with red arrows are replaced by the the one depicted with green arrows. Successful object-wise manipulation verifies that our method successfully disentangles the objects.

## A.9 EFFECT OF RANDOM SEEDS ON PERFORMANCE

We repeat our experiments on object disentanglement with 3 different seeds and report the values in Table 19. Our method shows comparable performance in object-centric tasks.

Table 19: Quantitative results on object disentanglement with 3 different runs for our model

| Method | CLEVREasy | | | CLEVR | | | | CLEVRTex | | |
|---|---|---|---|---|---|---|---|---|---|---|
| | Shape ($\uparrow$) | Color ($\uparrow$) | Position* ($\uparrow$) | Shape ($\uparrow$) | Color ($\uparrow$) | Material ($\uparrow$) | Position ($\downarrow$) | Shape ($\uparrow$) | Material ($\uparrow$) | Position ($\downarrow$) |
| SA | 72.25 | 72.33 | 44.08 | 79.4 | 91.30 | 93.18 | 0.064 | 30.44 | 7.890 | 0.482 |
| SLASH | 86.06 | 89.23 | 46.97 | 83.28 | 92.26 | 93.16 | 0.078 | 53.13 | 37.49 | 0.148 |
| LSD | **96.03** | **98.05** | 50.29 | **87.66** | 91.46 | **94.87** | 0.062 | 68.25 | 51.54 | 0.197 |
| L2C | 92.78 | 93.57 | 47.62 | 73.61 | 74.03 | 86.93 | 0.168 | **71.54** | **51.62** | **0.116** |
| **Ours** | 93.74±2.10 | 94.29±0.97 | 49.42±1.15 | 85.72±0.37 | **93.79±0.22** | 94.93±0.07 | **0.058±0.006** | 68.29±2.55 | 47.89±4.89 | 0.143±0.009 |

