# UNIFYING DISENTANGLED REPRESENTATION LEARNING WITH COMPOSITIONAL BIAS

## ABSTRACT

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

Our mixing strategy (Equation 2) shares connection to random permutation trick in FactorVAE (Kim & Mnih, 2018). FactorVAE explicitly forms a factorized posterior by randomly mixing each dimension of the latent representations among different images. However, FactorVAE assumes statistical independence on the latent space and propose to minimize KL divergence between a factorized posterior (*i.e.*, distribution of randomly mixed samples) and a aggregated posterior (distribution of sample). Such objective is specifically tailored for statistical independent factors of variation, make it trivial to apply this objective into disentangling other class of factors of variation, e.g., object. In contrast, we implement our inductive bias as a mixing strategy, we can without modifying the objective function, which will be introduced in next paragraph.

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

irrelevant to $\mathbf{z}^c$. This occurs because the encoder can collapse the posterior $p_\theta(\mathbf{z}|\mathbf{x})$ into a single mode [1], causing any two latents encoded by the encoder to maintain a small distance. In such cases, the decoder generating random realistic images for $\mathbf{z}^c$ may produce a small distance between the original and inverted latents, regardless of consistency. To address this issue, we instead minimize the *relative* distance between $\mathbf{z}^c$ and $\hat{\mathbf{z}}^c$, *i.e.*, the distance relative to negative samples, which are latents from other random images. This prevents the encoder from collapsing the posterior into a single mode, as $\mathbf{z}^c$ must not only match $\hat{\mathbf{z}}^c$ but also be distinguished from negative samples. Formally, we define the compositional consistency loss is defined as:

$$\mathcal{L}_{\text{Con}}(\theta) = -\log \frac{\exp(d(\hat{\mathbf{z}}^c, \mathbf{z}^c)/\tau)}{\sum_{i \in \{1, \ldots, B\}} \exp(d(\hat{\mathbf{z}}^c, \mathbf{z}^i)/\tau)}, \tag{6}$$

where $\tau$ and $d(\cdot)$ denote temperature and cosine similarity, respectively, and $B$ is a batch size. Note that we should consider the correspondence between $\mathbf{z}^c = \{\mathbf{z}_1^c, \ldots, \mathbf{z}_K^c\}$ and $\hat{\mathbf{z}}^c = \{\hat{\mathbf{z}}_1^c, \ldots, \hat{\mathbf{z}}_K^c\}$ to compute the cosine distance. This can be problematic for object disentanglement, as object-disentangled representations can have permuted orders due to our mixing strategy. In this case, we first apply the Sinkhorn-Knopp algorithm (Cuturi, 2013) to compute a soft assignment between $\mathbf{z}^c$ and $\hat{\mathbf{z}}^c$, then use the assignment-weighted sum of the distances to compute the loss.

**Overall Objectives** In summary, our framework is built upon an auto-encoding framework, which is implemented with denoising objective. To maximize the compositionality of composite images, we maximize the likelihood of the composite image $\mathbf{x}^c$ with pre-trained diffusion model $G_\psi$, and enforce compositional consistency to ensure resulting $\mathbf{x}^c$ consistent to $\mathbf{z}^c$. The overall objective is given as:

$$\mathcal{L}_{\text{Total}}(\theta, \phi) = \mathcal{L}_{\text{Diff}}(\theta, \phi) + \lambda_{\text{Prior}}\mathcal{L}_{\text{Prior}}(\theta) + \lambda_{\text{Con}}\mathcal{L}_{\text{Con}}(\theta), \tag{7}$$

where $\lambda_{\text{Prior}}$ and $\lambda_{\text{Con}}$ controls the relative importance of the objectives. Note that these objectives are not tailored specific to each factors of variation, but instead shared for both attribute and object disentanglement.

## 4 EXPERIMENT

**Implementation Details** We use the same encoder and decoder architectures as the baselines (Yang et al., 2023; Jung et al., 2024) for a fair comparison. Following the state-of-the-art methods in attribute (Yang et al., 2023) and object (Jiang et al., 2023) disentanglements, we employ a pre-trained VAE (Rombach et al., 2022) to represent an image as a latent feature and a latent diffusion model (Rombach et al., 2022) for the decoder $D_\phi$. Since the diffusion decoder operates on VAE features, we adjust our image encoder to take VAE features as an input. When generating the image $\mathbf{x}^c$ from $\mathbf{z}^c$, we iteratively decode images using only a few steps (1 to 4 steps) following DDIM (Song et al., 2020) to efficiently reduce the costly iterative decoding process. When back-propagate the gradient through $\mathbf{x}^c$, we truncate the gradient at the last iteration of decoding. Also, to ensure the reliable image generation from few-step decoding, we use a v-prediction objective when training the diffusion model (Salimans & Ho, 2022). For the generative prior $G_\psi$, we train an unconditional diffusion model on each training dataset from the scratch. More implementation details can be found in the Appendix A.4.

### 4.1 ATTRIBUTE DISENTANGLEMENT

**Datasets** We evaluate our method on three standard datasets in disentangled representation learning. Shapes3D (Kim & Mnih, 2018) consists of 3D shapes with 6 ground truth factors. Cars3D (Reed et al., 2015) is a dataset of 3D car models with 3 ground truth factors. MPI3D (Gondal et al., 2019) contains physical 3D objects with 7 factors of variation. All experiments are conducted at a 64x64 image resolution, following (Ren et al., 2022; Yang et al., 2023).

**Evaluation Metrics** We use two evaluation metrics: the FactorVAE (Kim & Mnih, 2018) score and the DCI (Eastwood & Williams, 2018) metric. The FactorVAE score measures disentanglement

---

[1] We measured the pairwise cosine distances between latents from different images and observed that all of them are signifiantly low