# OpenReview forum: "Unifying Disentangled Representation Learning with Compositional Bias"
_ICLR.cc/2025/Conference — Submitted to ICLR 2025_

### Official Review · Reviewer_bFYK · 2024-10-28

**Soundness:** 2
**Presentation:** 2
**Contribution:** 3
**Rating:** 6
**Confidence:** 4

**Summary:**

This work proposes a framework to learn disentangled representations of either attributes (e.g., an object's color or orientation) or distinct objects within a scene. The frameworks begins by encoding a pair of images using a VAE encoder. The embeddings generated are $k$ vectors that eventually will be the disentangled representations. At this stage a mixer samples some vectors from  image 1 and some vectors from image 2 generating the representation of a ``new’’ composed image. These new representation are then noised and denoised thanks to a diffusion model before going through the decoding stage of the VAE.  The mixing component can be adjusted according to the desired inductive bias. For attribute disentanglement, the model enforces mutual exclusivity by ensuring each latent vector is sampled from only one of the two images. In contrast, for object disentanglement, this exclusivity constraint is removed, allowing, for instance, the first latent vector to be sampled from both images.

The objective function is composed of three terms: (1) a latent denoising objective using a diffusion decoder (as in Jung et al., 2024); (2) a term to maximize the likelihood of the composed image, implemented as a diffusion loss, where the diffusion model is pre-trained for each task and then frozen; and (3) a consistency objective, which ensures that the latent representation $z$ of a given image and the latent representation re-encoded after decoding the reconstructed image from $z$ remain close. For this last term, the authors found that using an NCE-like objective, where each representation should be close to its counterpart and distant from other batch representations, outperformed simply minimizing cosine similarity.

The proposed method is evaluated against various baselines, datasets, and metrics for both attribute and object disentanglement, showing improved performance across the board.

**Strengths:**

The paper is easy to read. The proposed framework leverages and combines many techniques (such as diffusion models, SSL, optimal transport) in interesting way. The final framework is simple and from the reported results effective.

**Weaknesses:**

The main weaknesses of this paper are in the empirical evaluations. Specifically, some of the results reported do not match those previously published, a very common task used to assess object disentanglement  (unsupervised segmentation) is missing, none of the experiments are done on realistic or complex datasets (although recent state-of-art works do employ those kind of datasets). These are the main points to be discussed during the rebuttal. Fixing these could increase the soundness and the contribution scores, hence, also the final recommendation score. See below for more details on all these weaknesses.
- Results reported in this work about other baselines do not seem to match the original results reported by the respective original papers on the same tasks and datasets. For example LDS property prediction in the original paper shows much better accuracy (80.23% on Shape, compared to the one reported in this work for LSD which is only 68.25%, for comparison the proposed method accuracy is 70.90%). For the properties “material” and “shape” the differences is even higher.
- State of art works on object disentanglement consistently use unsupervised segmentation to assess the usefulness of the generated representations, however, these tests are missing from the current work. This is an important task because it shows a concrete application of these type of representations (and for ease of comparison given that all recent works use both unsupervised segmentation as well as property prediction).
- Both set of experiments (attributes and objects) lack of realistic or more complex datasets which state-of-the art have been using (in addition to some of the datasets used in this work). While it is not needed to have results on all of the following datasets, showing that the proposed method scales to the complexity of some of those datasets comparably to the state of art would make the contribution stronger. For example:
    - For the attribute disentanglement FactorVAE uses CelebA.
    - For Object centric Jung et al. 2024 use Super-CLEVR (multi-colored parts and textures), and MultiShapeNet (for realistic images), while other work such as Object Centric Slot Diffusion use the MOVi-C dataset (which contains complex objects and natural background),  MOVi-E datasets (which contains up to 23 objects per scene), FFHQ (high quality image of faces).

Other minor evaluations weaknesses:
- Attribute disentanglement results are reported with standard deviation (great!) but it is unclear on how many runs. Results for object disentanglement are provided without any standard deviation (but they should).

Minor Writing Comments. This writing suggestions are not critical but they would improve clarity and readability of the paper. No need to discuss them in rebuttal but they do need to be fixed and could increase the presentation score.
- I find the first part of the paper (until section 3) lacking important details that could easily be provided. For example:
    - The abstract is very dry, there is no mention of which are the “strong baselines”, nor which tasks this work was tested on, nor quantitative evaluation to show that the propose method “matches or exceeds” baselines. Consider adding more information.
    - From the abstract (and even the introduction and the beginning of section 3.1) it is not clear what “mix”, “compose”, “composition operator” mean. It could be concatenation, averaging, summing… it will only become clear much later but It would be great to provide more details if not in the abstract (ideal) at least in the introduction.
    - Still by the end of Section 2 there is no formal definition of “attribute” and “object”.  The first example of attributes is at page 4. Having these definitions would help the reader understanding the work much better since the beginning of the paper. From the examples at page 4 it seems that nose is an attribute and face an object but it could easily be argued that actually nose is an object in itself, or that face is an attribute of a bigger objet (human body). Again this highlight the need for a formal definition of attributes and objects.
- In Figure1 there is a concrete image example but it is not clear if it belongs to Attribute mixing or Object mixing. The “thing” being mixed is a cylinder and a ball so why is it linked both to attributes and objects? It would be clearer to provide an example for both. Note that everything becomes clearer once the whole paper has been read but the first time the reader reaches Figure 1 this could be a source of confusion.
- At page 6 the authors say “This occurs because the encoder can collapse the posterior pθ(z|x) into a single mode“. I know if this is an issue with posterior collapse.  If the encoder collapses the posterior, then the first loss ($L_{diff}$) should become high hence preventing the collapse. The problem seems to be related to the fact that the learnt encoding is sufficiently different (hence not collapsed) to keep $L_{diff}$ while what the authors want is not just $\hat{z} = z$ but also as different as possible with respect to other $z$s.
- Typo (?): “we can without modifying the objective function, which will be introduced in next paragraph.” It is not clear what is that “we can”.
- Typo: line 241 “an noised”.
- The following sentence is incomplete: “we adjust our image encoder to take VAE features as input”. Please clarify which kind of adjustments?
- “When back-propagate the gradient through xc, we truncate the gradient at the last iteration of decoding”. Why, it would be great to explain and motivate this choice.
- Typo in Line 310: “model on each training dataset from the scratch”. Should be “from scratch”.
- It would be great to explain how you understand which latent controls which factor. I believe there is a brief explanation in the appendix but it would be great if it could be explained in the main paper.
- In table 3 and some part of the appendix the loss term $L_con$ is called $L_cycle$. Please update it so that it is consistent throughout the paper.

**Questions:**

Please address the main weaknesses listed above. These are the most critical ones, I find the paper interesting but these weaknesses do need to be tackled, specifically:
A. Could you explain or correct the mismatch between your results and those previously reported?
B. Could you provide results on unsupervised segmentation tasks using the three typical metrics: Adjusted rand index for foreground objects (FG-ARI), mean intersection over union (mIoU), and mean best overlap (mBO) (see Jung et al 2024 as an example).
C. Could you provide results on at least a couple of the more complex datasets listed above (and for the tasks used in the state of art work mentioned).

Additionally these are more questions that are interesting to discuss.

D. The authors state at various points in the manuscript that previous methods use inductive biases specific to either attributes or objects, making them unsuitable for both simultaneously. For instance, in the statements, “Existing disentangled representation learning methods rely on inductive biases tailored for specific factors of variation (e.g., attributes or objects). However, these biases are incompatible with other classes of factors” and “Unlike previous methods, which introduce inductive biases tailored specifically to either attribute or object.”
However, the proposed method also requires a choice of mixing strategy tailored to either attributes or objects, which seems like an inductive bias itself, specific to one type of disentanglement. Could this advance choice also be considered a form of inductive bias that is specific to objects or attributes? Likewise, could state-of-the-art methods (e.g., Jung et al., 2024) also be modified to handle both attributes and objects? It’s unclear to me to what extent prior methods are fundamentally "unable" to address both types of disentanglement, as opposed their experiments being focused on of the the two tasks but potentially adaptable to the other in a way similar to how this proposed method can be adapted via choosing an appropriate mixing strategy.

E. In Section 2 the authors make the following comment “in object-centric scenes, the same objects can appear in different spatial locations, complicating the definition of independence metrics for object representations”. It would be great to show qualitatively in examples like Figure 2 what happens when the image contains 2 identical objects and one of them is added or removed from the image. Would the proposed framework work or would there be a confusion among those object. I say this in part out of curiosity and in part because in Figure 3 (right 3rd column for inserting) it seems the model is confusing two similar objects and is adding the one in the back rather then one in the front. Could you provide those qualitative examples (if not possible in the rebuttal then in a potential future version of the paper).

F. I could not find any detail (even in the appendix) about w(t). Could you please provide details about this function for both attribute and object tasks.

G. The authors mention that Jung et al. use a similar prior term but since they use the same diffusion model (as opposed to a pre-trained and frozen one) they are measuring $p(x^c|z^c)$ rather than $p(x^c)$.  I have two comments and questions about this:
1. Even when using a frozen diffusion model, wouldn’t the final decoded image be conditioned on $z^c$?
2. Regardless, I think this would be a good choice to compare. How does the current framework compare quantitatively to a similar framework that uses the term from Jung et al? Using Jung et al. solution would simplify the framework and reduce the need for training an extra model. Could you provide a comparison between these two options?

H. For the DCI metric the authors say “we perform PCA as post-processing on the representation before evaluation, following (Du et al., 2021; Yang et al., 2023)”. While I appreciate that this has been done before I wonder if it is a fair evaluation of disentanglement when it is applied only to some methods. Shouldn’t each vector $z_i$ be considered one of the “dimensions”. With PCA one is not measuring the disentanglement of each dimension but rather the disentanglement of a rotated version of the linear combination of the dimensions. This does not seem the same. Please help me understand why this makes sense and it is a fair evaluation, or if you agree with me that this is not a fair evaluation please compute and report the DCI score without PCA.

---

> ### Author Response · Authors · 2024-11-28
> **Official Response to Reviewer bFYK (1/5)**
>
> We thank the reviewer for valuable comments and suggestions. We have revised our paper and provide clarification following the reviewers’ writing suggestions. Below we respond to the individual questions.
>
> > **Q1.** Could you explain or correct the mismatch between your results and those previously reported?
>
> **A1**. We appreciate the valuable comments.
> The difference in reported metrics between baseline results is due to different experimental setups. In LSD's original experiments, the encoder $E_\theta$ directly encodes RGB images into slot representations, whereas in our experiments, encoder $E_\theta$ gets latent features obtained from a pretrained vae [1] as input. This difference comes from our effort to align input and output formats for all baselines to ensure fair and direct comparisons across methods. However, prior works on object-centric learning adopt diverse input and output formats (RGB image, VAE feature, DINO feature etc) across different models, which hinders direct comparison across the methods. To address this, we aligned both the input and output formats to latent features encoded by a pretrained vae [1] for all methods. Despite this modification to latent encoder for baselines, the results in Table 15,Table  16 and Figure 5 in Appendix A.7 consistently demonstrate that baselines still effectively learn object-centric representations under this standardized setup.
>
> To further address the concerns, we additionally trained our method using an RGB image encoder (the only modification) identical to LSD and L2C, and reported the property prediction results in the Table below. For comparison with both LSD and L2C, we followed the evaluation protocol of the L2C paper and compared our results to the values reported in the L2C paper. As shown in Table below, our method still achieves comparable results to state-of-the-art baselines, re-ensuring that our novel mixing strategy provides a robust inductive bias for learning object-centric representations. We hope our explanation along with the additional experiments showing comparable performance to baselines, addresses the reviewer’s concerns.
>
> | | Pos ($\downarrow$) | Shape ($\uparrow$) | Material($\uparrow$) |
> |------|:----------------:|:----------------:|:------------------:|
> | SLATE+ |0.1757|78.72|67.99|
> |  LSD |0.1563|85.07|_82.33_|
> |  L2C |_0.1044_ |**88.86**|**84.29**|
> | Ours |**0.1033**|_86.43_| 78.20|
>
>
> > **Q2.** Could you provide results on unsupervised segmentation tasks using FG-ARI, mIoU, mBO
>
> **A2**. We appreciate the insightful comments.
> Following the reviewer’s suggestion, we additionally evaluated unsupervised segmentation quality of pretrained encoder and reported it in Appendix A.7. Unlike slot-attention-based methods, our method does not have a built-in mechanism to directly express group memberships between pixels. Therefore, we trained a Spatial Broadcast Decoder [2] on top of frozen latent representations to predict explicit object masks for each latent representation. We train Spatial Broadcast Decoder with a reconstruction loss to recover the original image from frozen latents in an unsupervised manner, and it requires minimal training costs as the encoder remains frozen and the decoder is shallow. With this explicit object mask, we compare our method against two strong baselines in slot-attention-based works, LSD and L2C, on CLEVR and CLEVRTex. For a fair comparison, we evaluate the baselines using both slot-attention mask and object masks obtained by training a Spatial Broadcast Decoder on their frozen slot representations. The results are reported in Table 15,16 and Figure 5 in Appendix A.7.
>
> On the CLEVR , our method achieved the best mIoU, mBO scores and comparable FG-ARI. A high FG-ARI of our method indicates that each mask captures complete objects, confirming effective object disentanglement of our method. However, we observed that the background mask is split across multiple latents. This is because constant backgrounds in CLEVR does not affect compositional generation and thereby avoiding penalties from the compositional loss. Since the constant background carries minimal information, it does not impact the quality of object representations and compositionality, and we do not consider it a problem from an object-centric representation perspective. In the CLEVRTex, our methods outperforms both LSD and L2C for all three metrics. In Figure 5, we observed that our method consistently encodes complete objects into distinct latents, whereas LSD and L2C often split objects into multiple latents. Also, in contrast to CLEVR, as CLEVRTex has various background colors, our model successfully encodes all of the background information into a single latent.
> These experiments on unsupervised segmentation confirm that our pretrained encoder achieves effective object-wise disentanglement. Notably, our method outperforms baselines in object segmentation without relying on spatial clustering architectures like slot attention.

---

> ### Author Response · Authors · 2024-11-28
> **Official Response to Reviewer bFYK (2/5)**
>
> > **Q3.** Could you provide results on at least a couple of the more complex datasets?
>
> **A3**. We appreciate the valuable comments.
> Following the reviewer’s suggestion, we conducted additional experiments on CelebA-HQ for attribute disentanglement and MultiShapeNet for object disentanglement, respectively. Due to limited time in the rebuttal period, we conduct experiments on CelebA-HQ instead of CelebA, as we can employ a pretrained, off-the-shelf diffusion model ([3]) for the CelebA-HQ dataset. Experimental details and results are included in Appendix A.8.
>
> For the CelebA-HQ dataset, we use the attribute-mixing strategy to disentangle attribute factors. To verify the disentanglement of the learned representations, we swap each latent vector one by one between two images and present the resulting composition images in Figure 6 and 7. In the third column of each figure, we observe that while the source images lack bangs, the swapped images successfully generate bangs while preserving other attributes. Similarly, in the fourth and fifth columns, the facial expressions (e.g., smile) and skin tones of the target images are effectively transferred to the source images. These qualitative results demonstrate that our attribute-mixing strategy is capable of disentangling attribute factors, even in complex datasets like CelebA-HQ.
>
> We also validate our method on MSN dataset with object-wise manipulation and unsupervised segmentation. For the object-wise manipulation task, we encode pairs of images into $N=5$ object representations and exchange random object latents between the pairs to construct composite images. As shown in Figure 8, our method successfully performed object-level insertion and removal, demonstrating that each latent representation distinctly captures individual objects. This confirms that our approach effectively disentangles object representations within the latent space.
>
> For the unsupervised segmentation task, we measure FG-ARI, mIoU, mBO on object masks following common practices in object-centric literature. As our method does not have a built-in mechanism to directly express group memberships between pixels, we additionally train Spatial Broadcast Decoder on the frozen latent representations to predict explicit object masks for each latent representation (please refer to A2 and appendix A.8 for details). The results are reported in Table 17 in Appendix A.8.
> Among the competitive slot-attention based baselines, our method achieves second-best performances across all of three metrics. The high segmentation scores of L2C are mainly due to its slot-attention-based regularization term (see Equation 8 in the L2C paper), which explicitly encourages the slot masks to align with object shapes. Excluding L2C, our method outperforms rests of the baselines (LSD, SLATE) across all metrics, despite not employing a spatial clustering mechanism like slot attention. These results demonstrate the effectiveness of our framework in disentangling object representations in a complex dataset.

---

> ### Author Response · Authors · 2024-11-28
> **Official Response to Reviewer bFYK (3/5)**
>
> > **Q4.**  Could the proposed method's choice of mixing strategy be considered an inductive bias, similar to prior methods, and could state-of-the-art approaches also be adapted to disentangle both attributes and objects with modifications?
>
> **A4**. We appreciate the valuable comments.
> We clarify that our goal is not to eliminate inductive biases specific to factors of variation but to propose an **inductive bias that is compatible with disentangling multiple factors of variation (e.g., attributes and objects)**. Our method implements this bias in the form of a mixing strategy, which can flexibly disentangle either objects or attributes by adjusting only the mixing strategy, maintaining the same model parameterization and objective functions. In contrast, prior methods typically embed their inductive biases into the objective functions (e.g., information-theoretic objectives) or architectural designs (e.g., slot-attention encoder). It is non-trivial to adjust such objective functions or architectural designs to achieve both attributes and objects within a single framework and to the best of our knowledge, there are no successful methods for such adaptation.
>
> For instance, information-theoretic objectives proposed in attribute disentanglement often aims to reduce statistical dependencies among latent variables. However, extending this objective function to disentangle objects in object-centric scenes—where the number of objects varies or identical objects appear in different spatial locations—is non-trivial and no such work has been reported. On the other hand, object-centric learning relies heavily on architectural biases that enforce spatial exclusiveness, which cannot naturally handle spatially non-exclusive attributes (e.g., color, shape, or texture). It is also not trivial how we should design architectural biases to promote disentanglement of spatially non-exclusive attributes.
>
> In the same context, state-of-the-art methods like Jung et al. (2024) cannot simply disentangle both attributes and objects by simply modifying one component, such as the objective function, architecture, or mixing strategy; Specifically, (1) Even if the objective function or mixing strategy is adjusted, the slot-attention encoder inherently enforces spatial exclusiveness of factor, preventing disentanglement of spatially non-exclusive attributes. (2) Even when removing the slot-attention encoder, reuse of diffusion decoder as a generative prior hinders accurate likelihood estimation (see response to Q7) and absence of a compositional consistency loss further hinders effective attribute disentanglement (see Ablation Study in Table 3).
> We hope this explanation clarifies the clear contribution of our method and why prior methods have challenges in achieving both attribute and object disentanglements.
>
> > **Q5.** It would be great to show qualitatively in examples like Figure 2 what happens when the image contains 2 identical objects and one of them is added or removed from the image.
>
> **A5**. We appreciate the valuable comments.
> We would like to clarify that in Figure 3, the observed issue was not a confusion by our model but rather a misdrawn arrow for the target object to be inserted. We corrected the figure in the current version. Regarding the scenario of inserting or removing objects in a scene with multiple identical objects, we agree this is an intriguing question to explore. As our learning objective encourages the model to learn "compositional" concepts, we expect it to assign each object to distinct slots, even in scenes with multiple identical objects. We will conduct this experiment and include the results in the final version of the paper.
>
> > **Q6.** Details of w(t) in Equation 5.
>
> **A6**. We appreciate the valuable feedback. $w(t)$ is a timestep-dependent function derived by [4] and usually set to $\sigma^2_t=1-\bar\alpha_t$, where $\bar\alpha_t$ is a hyper-parameter controlling the noise schedules in the diffusion model [5]. We add this information in the main paper.

---

> ### Author Response · Authors · 2024-11-28
> **Official Response to Reviewer bFYK (4/5)**
>
> > **Q7.** What is the difference from the prior term in Jung et al. and how does the proposed framework compare quantitatively to Jung et al.'s approach?
>
>
> **A7**. We appreciate the thoughtful comments.
> To address the first question, our frozen diffusion model is an “unconditional” model that estimates $p(\mathbf{x}^c)$ and does not rely on $\mathbf{z}^c$ during estimating the likelihood. In contrast, L2C (Jung et al.) uses a conditional diffusion model that estimates  $p(\mathbf{x}^c|\mathbf{z}^c)$, and thereby likelihood estimation is directly conditioned on $\mathbf{z}^c$. Such distinction is crucial because estimation of $\log p_\psi(\mathbf{x}^c |\mathbf{z}^c)$ is inherently sensitive to the conditioning variable $\mathbf{z}^c$. In L2C, the conditional likelihood estimator $p_\psi(\mathbf{x}^c | \mathbf{z}^c)$ is learned using denoising losses with $\mathbf{z}$ encoded from individual images $\mathbf{x}$. When $\mathbf{z}^1$ and $\mathbf{z}^2$ are randomly composed to form $z^c$, the resulting $\mathbf{z}^c$ may become out-of-distribution (OOD) samples (i.e., unseen sample during training). Consequently, the estimation of $\log p(\mathbf{x}^c | \mathbf{z}^c)$ becomes inaccurate for OOD $\mathbf{z}^c$, and also there is no guarantee that maximizing $p(\mathbf{x}^c | \mathbf{z}^c)$ for OOD $\mathbf{z}^c$ yields realistic samples for $\mathbf{x}^c$. In contrast, our method employs an unconditional diffusion model pre-trained to estimate $p(\mathbf{x}^c)$ and it ensures robust estimation of $\log p(\mathbf{x}^c)$ regardless of $p(\mathbf{z}^c)$. Therefore, our prior term is more robust for maximizing the likelihood of $\mathbf{x}^c$ compared to that of L2C.
>
> Nevertheless, we agree that comparing our method to L2C ’s approach would strengthen the analysis. To address this, we conducted experiments on CLEVRTex replacing our prior loss with the one proposed in L2C. The results are presented in Table below. Those results clearly highlight the superior performance of our prior term over L2C. We hope this explanation and comparison clarify the advantages of our approach.
>
> |      | Shape ($\uparrow$) | Material($\uparrow$) | Position ($\downarrow$) |
> |------|:----------------:|:----------------:|:------------------:|
> | Ours |    **70.90**      |       **52.20**      |        **0.133**      |
> | Ours + L2C prior  |     54.58    |     27.18    |      0.165     |
>
> > **Q8.** Elaboration on why PCA is used when computing DCI and the resulting values are fair
>
> **A8**. We did not apply PCA to the entire latent representation. Instead, PCA was performed on each vector $\mathbf{z}_i$, and the principal component with the highest principal value was selected. This adaptation was necessary because DCI is typically computed dimension-wise, but we cannot treat vector representations as scalars. By extracting the dominant component for each $i\mathbf{z}_i$, we can compute DCI while maintaining a fair evaluation across methods. We hope this explanation resolves the concern.
>
> > **Q9.** Standard Deviation of experiments.
>
> **A9**. We appreciate the valuable comment.  In attribute disentanglement, we measure the performance for 10 different runs following DisDiff. Also, following the reviewer’s suggestion, we train our model with three different seeds and report the standard deviations in Table 19 of Appendix A.9. Due to the limited time for rebuttal period, this was conducted only for our method, but we will include standard deviations for all baselines as well in the final version of the paper.
>
> > **Q10.** Modify Figure 1 to prevent confusion with mixing examples
>
> **A10**. We appreciate the valuable suggestion. Following the suggestion, we modified the figure to clearly indicate that the illustrative example represents the object mixing strategy by adding a labeled arrow and explicitly clarifying it in the caption. While we first considered including examples for both attribute mixing and object mixing as suggested, the limited space in the figure made it challenging to accommodate both.

---

> ### Author Response · Authors · 2024-11-28
> **Official Response to Reviewer bFYK (5/5)**
>
> > **Q11.** Clarification on Compositional Consistency Loss
>
> **A11**. We acknowledge that our explanation may have been misleading. The reviewer’s understanding is correct. The issue we describe is not posterior collapse in the traditional sense. Rather, it refers to a scenario where reconstruction is successful ($(L_{\text{diff}}$ is very low), but $\mathbf{z}^i$ and $\mathbf{z}_j$ become close in the latent space for every data index $i, j$. In this situation, even if $\mathbf{x}^c$ generates an image irrelevant to $\mathbf{z}^c$, the penalty from $d(\mathbf{\hat z}^c, \mathbf{z}^c)$ remains low, reducing the effectiveness of the compositional consistency loss. To effectively penalize such cases, we introduce a contrastive term to ensure that $\mathbf{\hat z}^c$ remains close to $\mathbf{z}^c$ but as different as possible with respect to other negative samples $\mathbf{z}_j$.  We have updated the main paper to clarify this point and better explain how our method addresses this issue.
>
> > **Q12.** Motivation of gradient truncation trick
>
> **A12**. As diffusion models require iterative denoising steps to decode $\mathbf{z}^c$ into $\mathbf{x}^c$, it is computationally prohibitive to back-propagate the gradients through all of the denoising steps. To address this, we draw inspiration from recent works [6, 7] in diffusion-based optimization. These studies demonstrate that truncating gradients at the last iteration of the denoising process effectively balances computational feasibility and  optimization performance. Following this approach, we truncate the gradient at the last iteration of the denoising step to ensure efficient back-propagation of the gradient. We added this detail in the main paper.
>
> [1] Rombach et al., High-Resolution Image Synthesis with Latent Diffusion Models, in CVPR 22.
>
> [2] Watters et al., Spatial broadcast decoder: A simple architecture for learning disentangled representations in vaes, in Arxiv 19.
>
> [3] https://huggingface.co/CompVis/ldm-celebahq-256
>
> [4] Poole et al., Dreamfusion: Text-to-3d using 2d diffusion, in ICLR 2023.
>
> [5] Ho et al., Denoising Diffusion Probabilistic Models, in NeurIPS 2020.
>
> [6]  Clark et al., Directly fine-tuning diffusion models on differentiable rewards, ICLR24.
>
> [7] Aligning Text-to-Image Diffusion Models with Reward Backpropagation, ArXiv.

---

> > ### Comment · Reviewer_bFYK · 2024-11-28
> > **Post rebuttal comment**
> >
> > Thanks to these authors for their thorough rebuttal, for providing many clarifications and performing additional empirical evaluation. The information provided in the rebuttal alleviate my main concern about the empirical evaluation, I am going to reflect this in the scores and the rating.

---

> > > ### Author Response · Authors · 2024-12-02
> > > **Official Response to post rebuttal comment from Reviewer bFYK**
> > >
> > > Thank you very much for your reply and for all of your detailed comments, which certainly strengthened our work.
> > > We are happy to hear that we have addressed most of your concerns.
> > > If you have any further questions or concerns, please do not hesitate to let us know.
> > > Although the discussion period is nearing its end, we will do our best to address any remaining issues.

---

### Official Review · Reviewer_iDAj · 2024-11-01

**Soundness:** 2
**Presentation:** 3
**Contribution:** 2
**Rating:** 3
**Confidence:** 4

**Summary:**

The paper presents a framework for disentangled representation learning that targets both attribute—and object-based disentanglement within a single model. The authors formulate disentangled representation learning as maximizing the compositionality of randomly mixed latent representations of distinct images. The method uses a pre-trained diffusion model as an image generator and introduces an additional compositional consistency loss to encourage the composite images to remain faithful to the composite latent. The authors claim that their method can obtain superior performance in standard disentanglement benchmarks.

**Strengths:**

**Strengths:**

- The paper is relatively clear and easy to understand;

- The general idea of enforcing compositional consistency across mixed latent representations is fairly neat, and could possibly be extended to more challenging scenarios;

- The results seem to match or exceed some of the previous works on disentanglement benchmarks.

**Weaknesses:**

**Weaknesses:**

- The approach relies on a pre-trained diffusion model to ensure composite image realism, but this doesn’t guarantee alignment with the intended attribute or object combinations. As such, it is my understanding that this can compromise the interpretability and control of compositions in the general case, especially in more complex scenarios with subtle and/or hierarchical attribute/object relationships.
- There are no guarantees that the latent representations are identifiable under the current model, and by implication, neither are the compositions;
- The fixed mixing strategies, although appropriate for the simple cases studied, are quite rigid and likely would not adapt well to more complex scenarios in real data;
- The scope of the evaluation is limited to toy settings which is somewhat outdated given the recent progress in generative modelling.
- The writing is a little careless at times, there are numerous typos and/or grammatical issues some of which are mentioned below.

In my opinion, in its current state, this work largely sidesteps the key challenges in the area today, particularly the theoretical analysis of identifiability for latent representations and the development of scalable techniques that allow object-centric methods to be applied effectively in real-world settings. Therefore, I would encourage the authors to bolster their current contribution by tackling one of the two aforementioned challenges in the future.

**Typo corrections:**

line 34 "theoretically prove" \
line 46 "a unique object" \
line 70 "and verify" \
section 2 heading change to "Background" \
line 77 "incompatible with" \
line 97 "that render" \
line 107 "tailored specifically" \
line 122 "maximizing the likelihood" \
line 122 "disentangle attributes and objects" \
line 147 "to the type of" \
line 163 "While (Jung et al., 2024) rely" \
line 165 sentence needs rewriting for clarity \
line 167 "derive a specific" \
line 177 "of each factor" \
line 177 "derive a corresponding" \
line 188 "independent sampling of" \
line 190 "is equivalent" \
line 197 "always contains" \
paragraph starting at line 206 could do with rewriting for clarity \
line 216 "belong to the same" \
line 259 "While Jung et al. (2024) also maximize..." \
line 295 "to each factor of" \
line 307 "ensure reliable image generation" \
line 310 "from scratch" \
page 6 footnote "significantly" \

etc

**Questions:**

- What challenges do the authors anticipate in applying this model to real-world, complex datasets, and how might they address these?
- Could dynamic/learned mixing strategies replace fixed ones to improve flexibility in complex scenes?
- Have the authors thought about under which conditions their method can provide identifiability guarantees?

---

> ### Author Response · Authors · 2024-11-28
> **Official Response to Reviewer iDAj (1/3)**
>
> We thank the reviewer for valuable comments and suggestions. We have revised the paper to correct the typos and provide clarifications. Below, we respond to each of the individual questions.
>
> > **Q1.** Proposed method ensures composite image realism, but this doesn’t guarantee alignment with the intended attribute or object combinations.
>
> **A1**. We appreciate the valuable comment. As the reviewer pointed out, optimizing solely for realism of composite images may produce realistic images that do not align with the source attribute or object representations. We addressed such misalignment with compositional consistency loss. This loss explicitly enforces the accurate reconstruction of source latent representations from the generated composite images. By penalizing composite images that are realistic but fail to match the source latent representations, this loss ensures alignment between realistic composite images and intended latent compositions. We hope this clarifies our approach and addresses the reviewer’s concern.
>
> > **Q2.** There are no guarantees that the latent representations are identifiable under the current model, and by implication, neither are the compositions.
>
> **A2**. We agree with the reviewer that our method does not provide theoretical guarantees for the identifiability of latent representations or their compositions. While theoretical guarantees would certainly strengthen our work, **the primary focus of our paper is to demonstrate that the incompatible inductive biases traditionally used for disentangling attributes and objects can be replaced with a unified and compatible inductive bias in the form of a mixing strategy**. Furthermore, guaranteeing the identifiability of latent representations typically requires strong assumptions on the decoder function (e.g., [1,2]) or latent priors (e.g., [3,4,5]). As a result, much of the prior work in disentangled representation learning and object-centric learning often focus more on empirical performance rather than guaranteeing identifiability. Since our work prioritizes serving as a proof of concept, we also have adopted an empirical approach. Nevertheless, we agree that investigating when and how our framework can effectively learn object-centric representations and disentangle attributes within the context of identifiability theory would be a valuable direction for future research.
>
> > **Q3.** The fixed mixing strategies, although appropriate for the simple cases studied, are quite rigid and likely would not adapt well to more complex scenarios in real data.
>
> **A3**. We appreciate the valuable comment.
> We believe that the “fixed” mixing strategy could adapt to more complex scenarios as well. The role of mixing strategy is to define a specific form of compositional structure that the latent representation should satisfy. By controlling the mixing strategy, the model would discover different factors of variation aligned with the intended compositional structures. For instance, when an attribute mixing strategy is applied in real-world scenes, the model would learn to disentangle unique and composable factors that are always present in the scene (e.g., global lighting or style). Conversely, applying object mixing strategies to same scenes would guide the model to disentangle dynamically occurring object components within a scene. Therefore, we believe that the fixed mixing strategies are not a significant limitation and are applicable to both simple and complex scenarios.

---

> ### Author Response · Authors · 2024-11-28
> **Official Response to Reviewer iDAj (2/3)**
>
> > **Q4.** The scope of the evaluation is limited to toy settings which is somewhat outdated given the recent progress in generative modelling.
>
> **A4**. We appreciate the valuable comments. Following the reviewer’s suggestion, we conducted additional experiments on CelebA-HQ for attribute disentanglement and MultiShapeNet (MSN) for object disentanglement, respectively. Experimental details and results are included in Appendix A.8.
>
> For the CelebA-HQ dataset, we use the attribute-mixing strategy to disentangle attribute factors. To verify the disentanglement of the learned representations, we swap each latent vector one by one between two images and present the resulting composition images in Figure 6 and 7. In the third column of each figure, we observe that while the source images lack bangs, the swapped images successfully generate bangs while preserving other attributes. Similarly, in the fourth and fifth columns, the facial expressions (e.g., smile) and skin tones of the target images are effectively transferred to the source images. These qualitative results demonstrate that our attribute-mixing strategy is capable of disentangling attribute factors, even in complex datasets like CelebA-HQ.
>
> We also validate our method on MSN dataset with object-wise manipulation and unsupervised segmentation. For the object-wise manipulation task, we encode pairs of images into $N=5$ object representations and exchange random object latents between the pairs to construct composite images. As shown in Figure 8, our method successfully performed object-level insertion and removal, demonstrating that each latent representation distinctly captures individual objects. This confirms that our approach effectively disentangles object representations within the latent space.
>
> For the unsupervised segmentation task, we measure FG-ARI, mIoU, mBO on object masks following common practices in object-centric literature. As our method does not have a built-in mechanism to directly express group memberships between pixels, we additionally train Spatial Broadcast Decoder on the frozen latent representations to predict explicit object masks for each latent representation (please refer to A2 and appendix A.8 for details). The results are reported in Table 17 in Appendix A.8.
> Among the competitive slot-attention based baselines, our method achieves second-best performances across all of three metrics. The high segmentation scores of L2C are mainly due to its slot-attention-based regularization term (see Equation 8 in the L2C paper), which explicitly encourages the slot masks to align with object shapes. Excluding L2C, our method outperforms rests of the baselines (LSD, SLATE) across all metrics, despite not employing a spatial clustering mechanism like slot attention. These results demonstrate the effectiveness of our framework in disentangling object representations in a complex dataset.
>
> > **Q5.** What challenges do the authors anticipate in applying this model to real-world, complex datasets, and how might they address these?
>
> **A5**. One of the primary challenges in applying our model to real-world, complex datasets is the need for a diffusion model that can reliably estimate the likelihood of complex composite images. Fortunately, this challenge can be addressed by leveraging off-the-shelf diffusion models trained on large-scale datasets (e.g., Stable Diffusion) or fine-tuning them on the target dataset. Another practical challenge is that real-world datasets often consist of highly diverse and complex images, which may result in slower convergence if a randomly initialized encoder is used. To address this, employing a pretrained encoder (e.g., DINO) as commonly done in recent object-centric approaches, would likely improve training efficiency and representation quality.
> Lastly, our current mixing strategy is designed to disentangle attributes or objects separately, but it does not yet support discovering factors of variation with more complex compositional structures, such as jointly disentangling attributes and objects or handling hierarchical relationships. To address such cases, proper mixing strategies should be investigated to capture such intricate compositional structures. For example, to discover factors of variation with hierarchical structures, the mixing strategy could be adapted to enforce hierarchy-specific constraints, such as allowing exchanges only between nodes at the same level in the hierarchy. Exploring and embedding diverse compositional structures through advanced mixing strategies is an essential direction for future research, and we will investigate this in our future work.

---

> ### Author Response · Authors · 2024-11-28
> **Official Response to Reviewer iDAj (3/3)**
>
> > **Q6.** Could dynamic/learned mixing strategies replace fixed ones to improve flexibility in complex scenes?
>
> **A6**. We appreciate the insightful question. Yes, we believe dynamic or learned mixing strategies would be an interesting and effective extension of our work to flexibly discover various factors of variations with complex compositional structures. When we apply a fixed mixing strategy, the model will consistently discover factors of variation satisfying the specific compositional structure we embed through the mixing strategy. However, in real-world scenarios, the compositional structure of the factors may vary across scenes. In such cases, applying a mixing strategy adaptive to the given scene would help capturing scene-dependent factors of variations. Moreover, learning valid mixing strategies directly from data to uncover the underlying compositional structures would be another promising future direction.
>
> > **Q7.** Have the authors thought about under which conditions their method can provide identifiability guarantees?
>
> **A7**. Guaranteeing identifiability generally requires imposing strong restrictions on the class of decoders or the latent distribution. In recent object-centric literature, additive decoders have been widely adopted to guarantee identifiability [2, 6, 7]. For object disentanglement, our method could leverage an additive decoder as well and define compositionality using compositional contrast as proposed in [6, 7]. These steps would be the first step to provide identifiability guarantees for object representations. However, for factors of variation like attributes, which globally affect the image, the additive decoder and compositionality definitions may not be satisfied in general. In such cases, we believe that ensuring identifiability would require imposing specific structural constraints on the latent distribution, as explored in [3, 8]. Therefore, it is challenging to immediately identify a unified set of conditions that guarantee identifiability for both attributes and objects simultaneously. A promising direction would involve first establishing conditions for identifiability for attribute and object separately, and then develop a general theory to integrate these conditions.
>
> [1] Brady et al., Provably Learning Object-Centric Representations, in ICML 23.
>
> [2] Lachapelle et al., Additive Decoders for Latent Variables Identification and Cartesian-Product Extrapolation, in NeurIPS 23.
>
> [3] Hyv¨arinen et al., Nonlinear ica using auxiliary variables and generalized contrastive learning, in AISTATS, 19.
>
> [4] Khemakhem et al., Variational autoencoders and nonlinear ica: A unifying framework, in AISTATS, 20.
>
> [5] Khemakhem et al., Ice-beem: Identifiable conditional energy-based deep models based on nonlinear ica, in NeurIPS 20.
>
> [6] Brady et al., Provably Learning Object-Centric Representations, in ICML 2023.
>
> [7] Wiedemer et al., Provable Compositional Generalization for Object-Centric Learning, in ICLR 2024.
>
> [8] Lachappelle et al., Disentanglement via mechanism sparsity regularization: A new principle for nonlinear ICA, in Conference on Causal Learning and Reasoning, 2022

---

> > ### Comment · Reviewer_iDAj · 2024-12-02
> > **Thank you**
> >
> > I thank the authors for their detailed response. I agree with the authors that their approach serves primarily as ''a proof of concept'' and certainly has some merit. In the author's A3 response, a lot of assumptions are made about what the model "would" do in real-world settings by controlling the mixing strategy, but I'm not convinced that these are well-founded as it's not obvious what mixing strategy would enable unique and composable factors to be recovered. Since no new theoretical identifiability guarantees are given for their method and the results are not particularly strong relative to the baselines in my opinion, I remain sceptical that the contributions are significant enough to warrant a substantial score increase at this stage.

---

> > > ### Author Response · Authors · 2024-12-03
> > > **Official Response to post rebuttal comment from Reviewer iDAj**
> > >
> > > We sincerely thank the reviewer for the thoughtful comments and for recognizing that our work serves as a proof of concept and certainly has merit. We understand and agree with the reviewer that a theoretical foundation would strengthen our contribution. However, we respectfully believe that our current version still offers meaningful contributions to the field. We hope the reviewer will reconsider our work in light of these contributions.
> > >
> > >
> > > We would like to respectfully highlight that many state-of-the-art (SOTA) methods in this domain also do not provide theoretical identifiability guarantees [1,2,3,4,5,6]. Achieving identifiability often requires strong assumptions and thereby methods derived with guaranteed identifiability often suffer from limited performance. Instead, much of the existing work have focused on proposing practical and efficient necessary conditions for disentanglement, such as group constraints [7] or mutual information maximization [1,2,8]. Similarly, object-centric learning approaches, such as Slot Attention-based methods [4,5,6], promote object-level disentanglement via spatial exclusiveness but do not offer identifiability guarantees. In this context, our contribution lies in **proposing a novel, practical necessary condition—the mixing strategy—which enables disentanglement of both attributes and objects in a unified framework**.
> > >
> > >
> > > We also recognize the concern about the scalability of our mixing strategy in capturing diverse factors of variation in real-world settings. In Appendix A.8, the results demonstrate that our mixing strategy can achieve attribute and object disentanglement in more complex datasets such as Celeba-HQ and MultiShapeNet. Although the disentanglements were done on relatively simple factors, we note that existing state-of-the-art (SoTA) methods applied to the same datasets also primarily address basic factors such as global attributes (e.g., azimuth, skin tone, facial expression) [1,3] or object-level factors [4,6]. To the best of our knowledge, no prior work has yet addressed the disentanglement of complex (eg, hierarchical relations) or intricate factors of variation. This is a challenge for the field as a whole, not just our work.
> > >
> > >
> > > In this regard, we believe that our approach holds greater potential to generalize to complex factors of variation compared to prior methods. Previous frameworks are often designed to disentangle either only attributes or objects and lack inherent extensibility. In contrast, our unified framework, driven by the mixing strategy as a core inductive bias, is not limited to disentangling only attributes or objects. Instead, it can flexibly adapt based on the underlying compositional structure. For instance, hierarchical relationships among factors would be explicitly embedded though mixing strategy by allowing mixing only between nodes at the same hierarchical level, potentially enabling the discovery of such structures. We believe this flexibility represents an important contribution of our work, as our general framework could be extended to discover general factors of variations (eg, hierarchical factors) in complex scenarios.
> > >
> > >
> > > Lastly, regarding the comment that our results are “not particularly strong relative to the baselines,” we would like to emphasize that our method **uniquely** enables the disentanglement of both attributes and objects within a single, uniform framework. This capability, combined with results that outperform or remain comparable to SOTA methods, highlights the practical significance of our approach.
> > >
> > >
> > > We deeply appreciate the valuable feedback and the effort the reviewer have dedicated to reviewing our work. We hope the reviewer will consider reassessing our contributions in light of the broader context and the unique strengths of our method.
> > >
> > > References
> > >
> > > [1] Yang et al., Disdiff: Unsupervised disentanglement of diffusion probabilistic models, in NeurIPS 23.
> > >
> > > [2] Wang et al., Infodiffusion: Representation learning using information maximizing diffusion models, in ICML 23.
> > >
> > > [3] Ren et al., Learning disentangled representation by exploiting pretrained generative models: A contrastive learning view, in ICLR 20.
> > >
> > > [4] Jiang et al., Object-centric slot diffusion, in Neurips 23.
> > >
> > > [5] Wu et al., Slotdiffusion: Object-centric generative modeling with diffusion models, in Nuerips 23.
> > >
> > > [6] Jung et al., Learning to Compose: Improving Object Centric Learning by Injecting Compositionality, in ICLR 24.
> > >
> > > [7] Yang et al., Towards building a group-based unsupervised representation disentanglement framework, in ICLR 22.
> > >
> > > [8] Lin et al., Infogan-cr and modelcentrality: Self-supervised model training and selection for disentangling gans, in ICML 20.

---

### Official Review · Reviewer_cwzS · 2024-11-02

**Soundness:** 3
**Presentation:** 3
**Contribution:** 3
**Rating:** 6
**Confidence:** 2

**Summary:**

Note: I am not an expert on disentangled representation learning and know little/none of the related work.

The paper proposes an approach to learn a generative model for learning disentangled representations by maximizing the compositionality of representations. By mixing the representations of two images (given some constraints to make sure the results latent representations are valid) and maximizing the likelihood of the resulting composite images the model learns representations that can be disentangled on the object and attribute level. Experiments on synthetic datasets show that the model performs well in disentangling factors across several datasets both on the object and attribute level.

**Strengths:**

The paper addresses the learning of disentangled representations for both objects and attributes and makes use of a standard generative model for learning them. By introducing specific mixing strategies to combine latent representations of different images under given constraints the model is able to learn disentangled representations under a fairly simple framework.

The evaluation shows that the model learns better disentangled representations than the given baselines.

**Weaknesses:**

It seems like the approach is only useable if the practitioner already knows the underlying factors they want to disentangle, as the latent mixing strategies take this knowledge under account.
It's also not clear to me if this would translate to real-world datasets with more complicated distributions.
The experiments show results for either object disentanglement or attribute disentanglement but no experiments for joint object and attribute disentanglement.
All experiments are done on rather simple synthetic datasets.

**Questions:**

How would this generalize to more complex datasets where the exact factors of disentanglement might not be known. Does this scale to lots of disentangled factors (dozens or hundreds) or would that make the mixing strategies too complicated?

---

> ### Author Response · Authors · 2024-11-28
> **Official Response to Reviewer cwzS (1/2)**
>
> We thank the reviewer for valuable comments. Below, we respond to each of the individual questions.
>
> > **Q1.** It seems like the approach is only usable if the practitioner already knows the underlying factors they want to disentangle, as the latent mixing strategies take this knowledge under account.
>
> **A1**. We appreciate the valuable comment.
> While it is true that our current approach requires prior knowledge on target underlying factors to determine proper mixing strategy, we note that prior works also rely on such knowledge as they must select proper inductive bias (eg, either information-theoretic objectives for attributes or slot-attention encoder for objects).
> However, we believe that our mixing strategies would be still applicable even without exact information about underlying GT factors of variation in the data. The mixing strategy serves as a general framework for defining the compositional structure we expect our latents to follow, allowing the model to discover factors that align with a user-defined compositional structure. For instance, applying a fixed attribute mixing strategy encourages the model to disentangle globally consistent and combinable factors (e.g., global lighting or style) that are always present in the scene. Conversely, applying object mixing strategies to the same scenes helps the model disentangle dynamically occurring components within a scene, such as individual objects. In this way, the choice of mixing strategy flexibly determines the type of factors being disentangled.
>
> > **Q2.** The experiments show results for either object disentanglement or attribute disentanglement but no experiments for joint object and attribute disentanglement.
>
> **A2**. We appreciate the insightful comment.
> We agree that our current experiments focus on either object or attribute disentanglement separately, without addressing joint disentanglement. This choice was made because our **primary goal in this work was to demonstrate the capability of our novel inductive bias (i.e., compositional bias) to disentangle objects or attributes within a single framework**. By presenting separate experiments for object and attribute disentanglement, we aimed to clearly highlight the effectiveness of our method in achieving comparable performance on both tasks. We agree that exploring joint object and attribute disentanglement is an important direction that aligns with the broader goals of our framework. Although it is beyond the scope of the current work, we consider it an important direction for future research.

---

> ### Author Response · Authors · 2024-11-28
> **Official Response to Reviewer cwzS (2/2)**
>
> > **Q3.** Experiments are done on rather simple on rather simple synthetic datasets.
>
> **A3**. We appreciate the valuable comments.
> We conducted additional experiments on CelebA-HQ for attribute disentanglement and MultiShapeNet for object disentanglement, respectively. Experimental details and results are included in Appendix A.8.
>
> For the CelebA-HQ dataset, we use the attribute-mixing strategy to disentangle attribute factors. To verify the disentanglement of the learned representations, we swap each latent vector one by one between two images and present the resulting composition images in Figure 6 and 7. In the third column of each figure, we observe that while the source images lack bangs, the swapped images successfully generate bangs while preserving other attributes. Similarly, in the fourth and fifth columns, the facial expressions (e.g., smile) and skin tones of the target images are effectively transferred to the source images. These qualitative results demonstrate that our attribute-mixing strategy is capable of disentangling attribute factors, even in complex datasets like CelebA-HQ.
>
> We also validate our method on MSN dataset with object-wise manipulation and unsupervised segmentation. For the object-wise manipulation task, we encode pairs of images into $N=5$ object representations and exchange random object latents between the pairs to construct composite images. As shown in Figure 8, our method successfully performed object-level insertion and removal, demonstrating that each latent representation distinctly captures individual objects. This confirms that our approach effectively disentangles object representations within the latent space.
>
> For the unsupervised segmentation task, we measure FG-ARI, mIoU, mBO on object masks following common practices in object-centric literature. As our method does not have a built-in mechanism to directly express group memberships between pixels, we additionally train Spatial Broadcast Decoder on the frozen latent representations to predict explicit object masks for each latent representation (please refer to A2 and appendix A.8 for details). The results are reported in Table 17 in Appendix A.8.
> Among the competitive slot-attention based baselines, our method achieves second-best performances across all of three metrics. The high segmentation scores of L2C are mainly due to its slot-attention-based regularization term (see Equation 8 in the L2C paper), which explicitly encourages the slot masks to align with object shapes. Excluding L2C, our method outperforms rests of the baselines (LSD, SLATE) across all metrics, despite not employing a spatial clustering mechanism like slot attention. These results demonstrate the effectiveness of our framework in disentangling object representations in a complex dataset.
>
> > **Q4.** How would this generalize to more complex datasets where the exact factors of disentanglement might not be known?
>
> **A4**. We appreciate the reviewer’s thoughtful question.
> As discussed in our response to **Q1**, we believe that knowing the exact information of factors of variation is not strictly necessary for our approach. The latent representation would be learned according to the given mixing strategy, which guides specific compositional structure that the latent should satisfy. When there are an extremely large number of factors of variation in complex datasets, it would likely be infeasible to disentangle each exact factor. We hypothesize that the model would instead learn to group factors of variation into meaningful clusters.  If the factors of variation exhibit complex compositional structures, such as hierarchical relationships, extending our current mixing strategies would be necessary. For  instance, a hierarchical structure could be explicitly defined by allowing mixing only between nodes at the same level, which could enable the discovery of hierarchical factors of variation. Investigating mixing strategies for datasets with diverse and complex factors of variation is an important and promising direction for future research, and we aim to explore this in our future work.

---

### Official Review · Reviewer_XJvB · 2024-11-04

**Soundness:** 4
**Presentation:** 4
**Contribution:** 3
**Rating:** 8
**Confidence:** 3

**Summary:**

The paper attempts to tackle attribute and object disentanglement through the same mechanism as opposed to separate treatment by prior methods. Building on diffusion based decoding approaches that maximize compositionality, this paper lays emphasis on composing/mixing strategy of latents for object/attributes.

**Strengths:**

1. Addresses both attribute and object disentanglement by developing appropriate mixing strategy for latents. This is helpful to steer the field towards disentangling different types of factors of variation - eg properties of object and object themselves.
2. The paper gives an in depth analysis of the intricacies involved in optimizing for compositionality.
3. The paper is well written for the most part. There are appropriate visualizations in method and experiments that complement the text.

**Weaknesses:**

The impact of paper can be more by showing results on real world data

**Questions:**

Are there any further insights on the failure cases? Is it harder to compose attributes or objects?

---

> ### Author Response · Authors · 2024-11-28
> **Official Response to Reviewer XJvB**
>
> We thank the reviewer for valuable comments. Below, we respond to each of the individual questions.
>
> > **Q1.** The impact of paper can be more by showing results on real world data
>
> **A1.** We appreciate the valuable comments.
> Following the reviewer’s suggestion, we conducted additional experiments on CelebA-HQ for attribute disentanglement and MultiShapeNet for object disentanglement, respectively. Experimental details and results are included in Appendix A.8.
>
> For the CelebA-HQ dataset, we use the attribute-mixing strategy to disentangle attribute factors. To verify the disentanglement of the learned representations, we swap each latent vector one by one between two images and present the resulting composition images in Figure 6 and 7 in Appendix A.8. In the third column of each figure, we observe that while the source images lack bangs, the swapped images successfully generate bangs while preserving other attributes. Similarly, in the fourth and fifth columns, the facial expressions (e.g., smile) and skin tones of the target images are effectively transferred to the source images. These qualitative results demonstrate that our attribute-mixing strategy is capable of disentangling attribute factors, even in complex datasets like CelebA-HQ.
>
> We also validate our method on MSN dataset with object-wise manipulation and unsupervised segmentation. For the object-wise manipulation task, we encode pairs of images into $N=5$ object representations and exchange random object latents between the pairs to construct composite images. As shown in Figure 8, our method successfully performed object-level insertion and removal, demonstrating that each latent representation distinctly captures individual objects. This confirms that our approach effectively disentangles object representations within the latent space.
>
> For the unsupervised segmentation task, we measure FG-ARI, mIoU, mBO on object masks following common practices in object-centric literature. As our method does not have a built-in mechanism to directly express group memberships between pixels, we additionally train Spatial Broadcast Decoder on the frozen latent representations to predict explicit object masks for each latent representation (please refer to A2 and appendix A.8 for details). The results are reported in Table 17 in Appendix A.8.
> Among the competitive slot-attention based baselines, our method achieves second-best performances across all of three metrics. The high segmentation scores of L2C are mainly due to its slot-attention-based regularization term (see Equation 8 in the L2C paper), which explicitly encourages the slot masks to align with object shapes. Excluding L2C, our method outperforms rests of the baselines (LSD, SLATE) across all metrics, despite not employing a spatial clustering mechanism like slot attention. These results demonstrate the effectiveness of our framework in disentangling object representations in a complex dataset.
>
> > **Q2.** Are there any further insights on the failure cases? Is it harder to compose attributes or objects?
>
> **A2.** A failure case we observed occurs when a single latent encodes multiple factors of variation. However, this does not conflict with our compositional objective, as the objective remains satisfied even in such cases. We found that such issues can be mitigated by adjusting the weight of the compositional consistency loss. Specifically, the denominator of the compositional consistency loss includes a term that increases the distances between latents encoded from different images. This encourages the model to naturally encode distinct factors into separate latents, preventing the formation of empty or redundant latents in order to maximize the distance between each latent.
> In practice, we found that disentangling objects is a bit more challenging than disentangling attributes. This is due to spatial overlap between objects when they are randomly composed, which can occasionally result in unrealistic images. While this does not significantly impact overall training, it could cause slower convergence compared to attribute mixing, where mutually exclusive attributes are always composed without overlap.

---

### Official Review · Reviewer_c3eL · 2024-11-07

**Soundness:** 3
**Presentation:** 3
**Contribution:** 2
**Rating:** 6
**Confidence:** 3

**Summary:**

This paper investigates the learning of disentangled representations in particular the adaptation of existing frameworks to the learning of representations that can disentangle both attributes (e.g., color, texture, ...) and objects in a scene which authors claim prior work only tacked one of the other. The authors propose to leverage compositionality to learn disentangled representations. The setup includes pre-trained VAEs which provide representations that are then combined. The new representations serve as input to a diffusion-based decoder which is trained to reconstruct the composition of the original images. A pre-trained diffusion model is also used to enforce consistency between the input composite representations and the representation of the generated image. The method is tested for feature and object disentanglement on multiple synthetic datasets where is shows either superior or comparable performance to attribute or object disentanglement methods.

**Strengths:**

- presentation: the paper is polished, clear, and well-written
- relevance of the topics: learning models that disentangle sources of information whether attributes or objects without any prior knowledge about the type of sources but rather that rely on general prior information about the data structure like compositionality to enforce disentanglement is of great nterest to the community.

**Weaknesses:**

- complexity of the proposed approach leads to limited applicability and impact: the proposed approach requires the use of pretrained diffusion models to operate (i.e., to maximize the likelihood of composite images) and requires access to composite images to train the model.
- limited performance increase: while results show more consistent improvements for the **multi-seed** attribute disentanglement experiments, the gains are less consistent across metrics for the **single-seed** object disentanglement experiment.


Minor:
- theta should be a subscript in line 187
- typo line 212, 281, 310
- error in figure 1: z3 should be blue instead of orange
- line 227: figure 1 above

- Not sure I am getting lines 210-213

**Questions:**

- can authors elaborate on why the maximum likelihood is needed despite already enforcing low reconstruction error ?

---

> ### Author Response · Authors · 2024-11-28
> **Official Response to Reviewer c3eL**
>
> We thank the reviewer for valuable comments and suggestions. We have revised the paper to correct the typos and provide clarifications. Below, we respond to each of the individual questions.
> > **Q1.** Complexity of the proposed approach leads to limited applicability and impact: the proposed approach requires the use of pretrained diffusion models to operate and requires access to composite images to train the model.
>
> **A1.** We appreciate the reviewer’s comment and acknowledge the point about the need for pretrained diffusion models.
> Fortunately, with recent advancements in diffusion models, large-scale, off-the-shelf, pretrained models (e.g., Stable Diffusion) are now readily accessible.
> By leveraging these models, we can avoid the additional computational burden of training diffusion models from scratch when working with real-world datasets. Furthermore, generating composite images does not introduce additional learnable parameters, as we reuse the diffusion decoder trained with an auto-encoding loss. Therefore, while our method introduces some computational costs for generating and validating composite images, we do not need additional learnable components, preserving the overall applicability.
> > **Q2.** Limited performance increase in single-seed object disentanglement experiment.
>
> **A2.** We appreciate the valuable comment.
> To examine the robustness of our method in object disentanglement, we trained our model using three different random seeds and report the standard deviations in Table 19 in Appendix A.9. Due to the limited time budget during the rebuttal period, we conducted this analysis only for our method but we will include standard deviations for all baseline methods as well in the final version of the paper to provide a comprehensive comparison.
> Regarding the limited performance increase in object disentanglement experiment, we would like to emphasize that the primary goal of our work is **not to achieve state-of-the-art performance in both attribute and object disentanglement**, but **to propose a unified framework capable of disentangling both attributes and objects.** While our method shows comparable performances to baselines in object disentanglement experiments, it is the only approach among the competitors that can successfully disentangles both attributes and objects within a single framework. We believe this novel capability of our method brings a meaningful contribution to the field.
>
> > **Q3.** Can authors elaborate on why the maximum likelihood is needed despite already enforcing low reconstruction error?
>
> **A3.** We appreciate the reviewer’s comment. Minimizing the reconstruction error ensures that each latent representation is informative for a given image, but it does not inherently guarantee compositionality or realism in the generated composite images. To address this, we need an additional mechanism to encourage composite images to be realistic. Since ground-truth images for composite images are not available, we employ a pre-trained diffusion model to estimate the likelihood of composite images. By maximizing the likelihood, the model is guided to produce realistic composite images, thereby facilitating the learning of meaningful compositional representations.

---

> > ### Comment · Reviewer_c3eL · 2024-11-30
> >
> > Thank you for your clarification and additional results. The addition of seeds to the object disentanglement experiments confirmed that the proposed method performs on par or slightly worse than baselines.
> > While the weaknesses were all discussed by the authors, I don't believe the discussion motivates a score increase.

---

> > > ### Author Response · Authors · 2024-12-03
> > > **Official Response to post rebuttal comment from Reviewer c3eL**
> > >
> > > Thank you very much for your reply. We deeply appreciate your support and thoughtful feedback, which certainly strengthened our work.

---

### Meta-Review · Area_Chair_x6wV · 2024-12-04

**Metareview:**

This paper unifies the disentanglement of objects and factors of variation within a single framework. I think this is an important direction that deserves more investigation, and this paper is a step in the right direction. The method combines different architectural components and training methodologies creatively and effectively, achieving reasonable results. After discussions with the reviewers, I decided to reject the paper for the following reasons:

1. the results are not clearly better than competitors. However, it should be acknowledged that the authors did a good job, improving the complexity and realism of their experiments during the rebuttal period. **Suggestion:** I would encourage the authors to up their baseline choices, including recent works on object-centric disentanglement (e.g., neural systematic binder). I think clearly showing the benefits of the framework is key.

2. there is a lot of theory around both disentangled representations and object-centric learning. This is completely ignored in the paper, but I think it should be central to the benefits of the framework. This is particularly relevant as the paper also relies on weak supervision signals, but it is not clear how they differ from or if they are stronger or weaker than what is already used in the vast literature on identifiable disentangled representations.

3. I think that the writing needs to be made much more precise. I do not understand the difference between "general factors of variation" and "nuanced and intricate factors of variation." These should be defined in the paper.

**Additional Comments On Reviewer Discussion:**

The reviewers discussed with the authors and I also exchanged questions with them. I also read the paper myself, because I am interested in this area and the paper was otherwise borderline.

---

### Decision · Program_Chairs · 2025-01-22

Reject